# TorchGDM: A GPU-Accelerated Python Toolkit for Multi-Scale Electromagnetic Scattering with Automatic Differentiation

Sofia Ponomareva,[1,2] Adelin Patoux,[3] Clément Majorel,[3] Antoine Azéma,[1,2]
Aurélien Cuche,[2] Christian Girard,[2] Arnaud Arbouet,[4] and Peter R. Wiecha[1,*]

[1]*LAAS-CNRS, Université de Toulouse, Toulouse, France*
[2]*CEMES-CNRS, Université de Toulouse, Toulouse, France*
[3]*CRHEA-CNRS, Université Côte dAzur, Sophia Antipolis, France*
[4]*CNRS, Université Rennes, DYNACOM (Dynamical Control of Materials Laboratory) - IRL 2015,
The University of Tokyo, 7-3-1 Hongo, Tokyo, 113-0033, Japan*

We present "torchGDM", a numerical framework for nano-optical simulations based on the Green's Dyadic Method (GDM). This toolkit combines a hybrid approach, allowing for both fully discretized nano-structures and structures approximated by sets of effective electric and magnetic dipoles. It supports simulations in three dimensions and for infinitely long, two-dimensional structures. This capability is particularly suited for multi-scale modeling, enabling accurate near-field calculations within or around a discretized structure embedded in a complex environment of scatterers represented by effective models. Importantly, torchGDM is entirely implemented in PyTorch, a well-optimized and GPU-enabled automatic differentiation framework. This allows for the efficient calculation of exact derivatives of any simulated observable with respect to various inputs, including positions, wavelengths or permittivity, but also intermediate parameters like Green's tensor components, which can be interesting for physics informed deep learning applications. We anticipate that this toolkit will be valuable for applications merging nano-photonics and machine learning, as well as for solving nano-photonic optimization and inverse problems, such as the global design and characterization of metasurfaces, where optical interactions between structures are critical.

**Keywords:** Green's Dyadic Method, multi-scale photonics, automatic differentiation, nano-optics

## I. INTRODUCTION

Although Maxwell's equations are known for more than one and a half centuries, nano-optics simulations are notoriously difficult to solve and often require important computational resources. This is especially true for multi-scale simulations, which require accurate results at different length scales.[1]

Gustav Mie expanded the fields scattered by spherical particles in multipolar contributions using spherical harmonics.[2] Such multipole decomposition of electromagnetic fields is today also frequently used for particles of non-spherical geometries,[3–6] which is the theoretical foundation of the T-matrix method.[7–12] Likewise, effective electric (ED) and magnetic dipole (MD) point-polarizabilities are a commonly used semi-analytical model, appealing thanks to its formal simplicity. It can be used to describe large assemblies of nano-scatterers, in cases where the truncation of the multipole expansion at the dipole order is valid.[13–16] However, these methods allow a description only outside of a circumscribing sphere around non-spherical particles. Earlier attempts to overcome this problem only offered partial solutions to the problem.[17]

Therefore, semi-analytical methods like the T-matrix approach imply limitations for certain use-cases. Examples are dense arrays of non-spherical particles where the circumscribing spheres overlap, or scenarios with illumination by local emitters like fluorescent molecules or quantum dots close to the particles or inside the circumscribing sphere, or when internal fields are required. For perfect periodic systems, local emitters also pose a challenge. A phased-array scanning method can be used to decompose the dipole field in periodic contributions, however this technique is algorithmically and computationally very demanding.[18–20]

Recently proposed methods like the global polarizability matrix method ("GPM")[21] or the topological skeleton of multiple-multipoles[22] distribute several multipolar sources across the particle. However, at least inside the volume of the nanostructures such models remain approximative.

To overcome these limitations, we propose to combine the semi-analytical "GPM" method with a volume integral approach, the Green's Dyadic Method (GDM).[23,24] In our description, particles can be either represented by sets of effective electric and magnetic point dipole pairs (GPM),[25] or as a volume discretization (in 2D: surface discretization). The latter is strongly inspired by our former GDM implementation "pyGDM".[26,27] Dipole emitters can be used as illumination source, and the photonic local density of states (LDOS) and Green's tensors in complex environments can be calculated.[28,29]

The first peculiarity of our "TorchGDM" toolkit is that both types of particle representations (effective polarizability models and volume discretizations) can be mixed in a single multi-scale simulation. The second particularity is that torchGDM is fully implemented in the automatic differentiation (autodiff or "AD") framework *PyTorch*, making any simulation entirely differentiable with respect to any input (like positions, wavelength, permittivity, ...) or intermediate variables (e.g. Green's tensor components).

AD can be seen as a generalization of the adjoint method,[30–36] and promises great benefits compared to conventional global optimization methods for solving nano-photonics inverse problems such as design tasks, field retrieval or mode extraction.[37–41] In particular, we foresee applications for large-scale inverse problems with complex optical behavior such as the design of metasurfaces or complex media.[42–46]

## A. Similar software

Before we describe torchGDM in detail, we want to give a non-exhaustive selection of electrodynamics simulation methods, briefly discuss their pros and cons, and provide a selection of openly available toolkits.

### 1. Surface integral method

Best for single particle simulations, can handle complex shapes. Possibility to implement solver schemes with guaranteed convergence. Usually too computationally heavy for many-particle simulations.

- Null-field method, NFM-DS (fortran)[47]. Useful for calculating T-matrices of non-spherical particles.

- MNPBEM (matlab)[48–50]. Feature rich toolbox, strong focus on fast-electron simulations (electron energy loss spectroscopy, cathodoluminescence).

- nanobem (matlab)[51]. Very stable tool for simulation of individual nano-particles. Supports stratified environment. Convergence is guaranteed through the Galerkin method.

- NGSolve (c++, python)[52]. A generic framework for PDE solving. Includes a boundary element package for solving the Helmholtz equation. Also supports shape differentiation[53].

### 2. Discrete Dipole Approximation - iterative solver

Good for single particle simulations. Convergence is not guaranteed and it can show unexpected convergence problems. It is possible to improve the accuracy for instance using filtering techniques[54]. Improvements have also been suggested for specific applications such as high aspect ratio particles[55], high index dielectric materials[56], or periodic structures[57]. With the iterative solver, each incident field configuration needs an individual simulation run, therefore it can be expensive for multiple illuminations.

- DDSCAT (fortran)[58,59]. Well tested, established software.

- IFDDA (fortran, C++)[60]. Supports fully retarded multilayer environment and periodic structures. Several tools for microscopy applications. Graphical user interface.

- ADDA (C, GPU support)[61,62]. Very efficient implementation. Various extensions like substrate simulations or GPU acceleration.

### 3. Discrete Dipole Approximation - direct solver

Good for single particle. In principle, same as iterative DDA, but limited to smaller particle sizes and reduced accuracy, due to discretization limitations. Very well suited for multi-illumination simulations, because the inverted coupling system can be reused. Simple to implement.

- pyGDM (python, GPU support)[26,27]. Only volume discretization and electric-electric response supported.

- CEMD (Julia, GPU support)[63]. Coupled electric-electric and magnetic-magnetic polarizabilities, however no mixed terms supported. Volume discretization possible, but without self-terms, therefore less stable.

### 4. T-matrix method

Best for large multi particle simulations. Only solves the coupling between particles. Single particle t-matrices need to be calculated beforehand. Particles must not be closer than their circumscribing spheres. To the best of our knowledge, so far no automatic differentiation implementation exists.

- Treams (python, cython)[64,65]. Modern and clean toolkit with focus on 1D, 2D and 3D periodic structures.

- Smuthi (python, GPU support)[10]. Feature rich and accessible python API, focus on simulations in layered environment.

### 5. RCWA

Fourier modal method. Best for periodic structures, possibly in layered systems. Not ideal for single particles due to implicit periodic assumption. Very efficient in 1D and 2D, as well as for simple 3D structures. Expensive for complex 3D structures. Can show convergence problems in structures with complex interfaces (like dielectric/metal).

- Reticolo (matlab)[66]. Feature rich, well developed, extensively tested.

- MEENT (python)[67]. Support for automatic differentiation via jax and pytorch backends.

- FMMAX (python)[68]. Support for automatic differentiation via jax.

- nannos (python)[69]. Support for automatic differentiation via various backends.

### 6. FDTD

Very general method. Well suited for single particles and periodic simulations, as well as for larger models thanks to a friendly scaling behavior. Evanescent fields or strong gradients require fine meshes and can pose convergence problems. A simple FDTD code is easy to implement.

- MEEP (python)[70]. The open source FDTD reference toolkit. Very versatile and complete.

- FDTDX (python)[71]. Written in JAX, it supports (multi) GPU acceleration and automatic differentiation.

### 7. *This work*

The here presented tool "TorchGDM" implements a hybrid coupled dipole method that allows to combine volume discretizations as in the discrete dipole approximation (in the following we will call it the Green's Dyadic Method, GDM[24]), with effective point source models, like effective electric/magnetic dipole polarizabilities[25] and global polarizability matrices (GPMs).[21] GPMs allow efficient and accurate modelling of large particles and calculation of near-fields also inside the circumscribing sphere of non-spherical particles. TorchGDM supports 3D as well as 2D simulations. With the exception of an interface to the "treams" T-Matrix toolkit, the code is entirely written using "PyTorch" as backend, a popular automatic differentiation framework from the deep learning community. This allows to calculate derivatives of any possible observable with respect to arbitrary input or intermediate parameters. Finally, PyTorch offers full GPU support, simulations can therefore be run on accelerator hardware with no additional effort.

The most important capabilities are:

- 2D and 3D simulations

- Volume discretized particle models

- Effective dipoles based particle models

- Overlapping circumscribing spheres of effective models

- Combining particle types for multi-scale simulations

- Various far-field observables (complex far-field, total and differential cross sections, radiation patterns, exact multipole decomposition)

- Various near-field observables (electric and magnetic near-fields, fields inside discretized structures, chirality, Poynting vector, energy flux, field gradients, LDOS, Green's tensors, multipole decomposition)

- Automatic differentiation of any simulation calculation (with very few exceptions, see section II G)

- Various visualization tools

- Interfaces to external T-Matrix and Mie tool ("treams") for effective model creation (not AD compatible)

We believe that torchGDM will be particularly useful at medium size or multi-scale scattering problems, in cases where derivatives are required. Examples include design tasks like metasurface design, for instance when more than next-neighbor interactions are relevant for an accurate description. AD may also be useful in problems where a numerical model is to be fitted to experimental results. The differentiability can also be used to train neural networks within the simulation description, for example to learn effective Green's tensors or structure models.

In the future, further automatic differentiable components may be added. Possible developments include differentiable Mie theory or T-Matrix support, or wavefront propagation via the angular spectrum method. Layered media or periodic structures would also be interesting future developments, that could be added through Green's tensors as additional environment classes without the need to modify the main code itself. Finally, the full inversion scheme currently limits torchGDM to roughly 10000 coupled dipoles. This could be improved through iterative solver schemes. This could also be modularly implemented as alternative linear system class.

## II. FORMALISM AND IMPLEMENTATION

In the following we use cgs units, as well as a compact notation for electromagnetic fields, similar to the one used by Sersic et al.[72] The dependency on the frequency $\omega$ is omitted for the sake of readability. Here we start from the electromagnetic Lippmann-Schwinger equation, that can be derived from the wave equations for the electric and magnetic field:[27,73]

$$\mathbf{F}(\mathbf{r}) = \mathbf{F}_0(\mathbf{r}) + \int_V \mathbb{G}(\mathbf{r}, \mathbf{r}')\boldsymbol{\chi}(\mathbf{r}')\mathbf{F}(\mathbf{r}')\mathrm{d}\mathbf{r}'. \quad (1)$$

$\mathbf{F}(\mathbf{r}) = (E_x, E_y, E_z, H_x, H_y, H_z)^T$ is a 6-vector containing the complex, total electric and magnetic field at $\mathbf{r}$, $\mathbf{F}_0$ is the incident field in same notation. $\boldsymbol{\chi}$ is a $(6 \times 6)$ tensor containing the local electric and magnetic susceptibilities

$$\boldsymbol{\chi}(\mathbf{r}) = \begin{pmatrix} \boldsymbol{\chi}_{ee}(\mathbf{r}) & \boldsymbol{\chi}_{em}(\mathbf{r}) \\ \boldsymbol{\chi}_{me}(\mathbf{r}) & \boldsymbol{\chi}_{mm}(\mathbf{r}) \end{pmatrix}. \quad (2)$$

Note that in natural materials, typically only $\boldsymbol{\chi}_{ee}(\mathbf{r}) \neq 0$. Finally, $\mathbb{G}$ is the $(6 \times 6)$-Green's Dyad describing light propagation from a point source in the environment:

$$\begin{pmatrix} \mathbf{E}(\mathbf{r}) \\ \mathbf{H}(\mathbf{r}) \end{pmatrix} = \mathbb{G}(\mathbf{r}, \mathbf{r}') \begin{pmatrix} \mathbf{p}(\mathbf{r}') \\ \mathbf{m}(\mathbf{r}') \end{pmatrix}. \quad (3)$$

It is composed of the electric, magnetic and mixed Green's tensors (for the vacuum solutions of the Green's tensors in 3d and 2d, see Ref.[27]):

$$\mathbb{G}(\mathbf{r}, \mathbf{r}') = \begin{pmatrix} \mathbf{G}^{\mathrm{Ep}}(\mathbf{r}, \mathbf{r}') & \mathbf{G}^{\mathrm{Em}}(\mathbf{r}, \mathbf{r}') \\ \mathbf{G}^{\mathrm{Hp}}(\mathbf{r}, \mathbf{r}') & \mathbf{G}^{\mathrm{Hm}}(\mathbf{r}, \mathbf{r}') \end{pmatrix}. \quad (4)$$

The superscripts indicate the type of field (electric "E" or magnetic "H") at location $\mathbf{r}$, and the type of dipole source (electric "p" or magnetic "m") at location $\mathbf{r}'$, linked by the respective sub-tensor.

Through volume discretization on a regular grid into $N$ mesh cells, we obtain a linear system of $6N \times 6N$ equations:

$$\mathbf{F}(\mathbf{r}_i) = \mathbf{F}_0(\mathbf{r}_i) + \sum_{j=1}^{N} \mathbb{G}(\mathbf{r}_i, \mathbf{r}_j)\boldsymbol{\alpha}_j\mathbf{F}(\mathbf{r}_j). \quad (5)$$

## (a) reference system

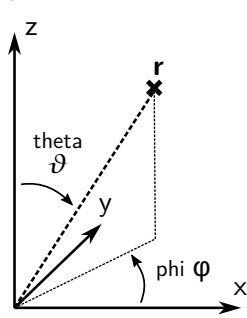

## (b) effective GPM models

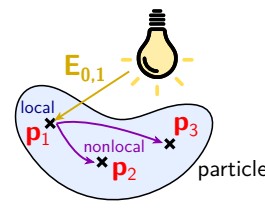

$$\boldsymbol{\alpha}_{\text{GPM}} = \begin{pmatrix} \alpha_{1\to1} & \alpha_{2\to1} & \alpha_{3\to1} \\ \alpha_{1\to2} & \alpha_{2\to2} & \alpha_{3\to2} \\ \alpha_{1\to3} & \alpha_{2\to3} & \alpha_{3\to3} \end{pmatrix}$$

## (c) 3D environment structure types

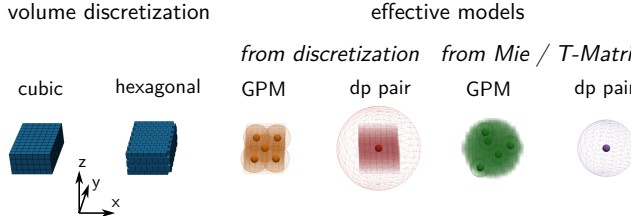

## (d) 2D environment structure types

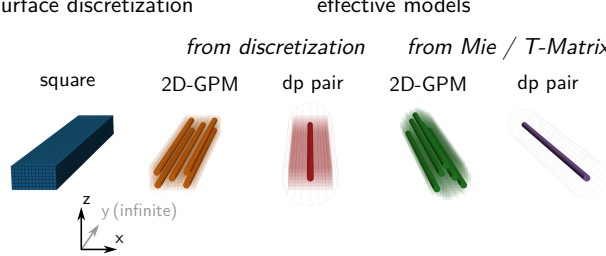

FIG. 1. (a) Reference system and angular conventions in torchGDM. (b) Sketch of the global polarizability matrix (GPM) effective model formalism.[21] A GPM describes a particle through a set of local (dark blue) and non-local polarizabilities. This means, the incident field at *one specific* polarizability location induces electric and magnetic dipole moments at *all* polarizability locations in the GPM. (c-d) available structure types for (c) 3D and (d) 2D simulations. In addition to discretized structures, effective dipole models are supported that may contain several non-local dipoles (global polarizability matrix, "GPM" (b)), or a single pair of an electric and a magnetic dipole ("dp pair"). Effective models can be extracted from discretized structures, T-Matrices or from Mie theory. Note that the Mie and T-Matrix extractions are not AD compatible.

Equation (5) introduces effective ($6 \times 6$) dipolar point-polarizabilities $\boldsymbol{\alpha}_i$ for each position $\mathbf{r}_i$. These polarizabilities are tensorial proportionality coefficients between the electric and magnetic fields and the locally induced electric and magnetic dipole moments $\mathbf{p}$ and $\mathbf{m}$ at location $\mathbf{r}_i$:

$$\begin{pmatrix} \mathbf{p} \\ \mathbf{m} \end{pmatrix} = \begin{pmatrix} \boldsymbol{\alpha}_{pE} & \boldsymbol{\alpha}_{pH} \\ \boldsymbol{\alpha}_{mE} & \boldsymbol{\alpha}_{mH} \end{pmatrix} \begin{pmatrix} \mathbf{E} \\ \mathbf{H} \end{pmatrix} = \boldsymbol{\alpha}\,\mathbf{F}. \qquad (6)$$

*Numerical solution* Equation (5) can be solved numerically with standard inversion methods like LU decomposition, by reformulating it into matrix form:[26]

$$\mathbf{F}_0(\mathbf{r}_i) = \sum_{j=1}^{N} \mathbf{F}(\mathbf{r}_j) \Big[ \delta_{ij}\mathbf{I} - \mathbb{G}(\mathbf{r}_i, \mathbf{r}_j)\boldsymbol{\alpha}_j\mathbf{F}(\mathbf{r}_j) \Big], \qquad (7)$$

where $\mathbf{I}$ is the identity tensor.

Note that automatic differentiation of the LU decomposition algorithm is numerically stable, which has also been demonstrated in the context of the coupled dipoles formalism in recent literature.[74,75]

### A. Volume discretization polarizabilities

TorchGDM treats two cases of polarizabilities: polarizabilities of mesh cells within a volume discretization and effective polarizabilities describing an entire nanostructure.

Natural materials respond only to the electric field of light,[76] therefore in the case of discretization mesh cells, only $\boldsymbol{\alpha}_{pE} \neq 0$. The polarizabilities are proportional to the volume (or in 2D simulations the surface) of the mesh cell and to the electric susceptibility of the material.[24,77] These polarizabilities are furthermore normalized by an additional factor that depends on the grid type. For a 3D discretization using cubic mesh cells of volume $d^3$, the polarizability writes

$$\boldsymbol{\alpha}_{pE} = \frac{\boldsymbol{\chi}_{ee} d^3}{4\pi}, \qquad (8)$$

where $d$ is the effective side length of a volume (or surface) element. So far, torchgdm supports cubic (3D), hexagonal (3D) and square (2D) lattice discretization (see figure 1c-d). For details, see Ref.[27]

*Self-terms* To treat the divergence of the Green's tensor at $\mathbf{r} = \mathbf{r}'$ (in a discretization when $i = j$), $\mathbb{G}$ needs to be replaced in Eq. (5) by so-called "self-terms"

$$\mathbb{G}(\mathbf{r},\mathbf{r}) = \mathbb{S} = \begin{pmatrix} \mathbf{S}^{\text{Ep}} & \mathbf{S}^{\text{Em}} \\ \mathbf{S}^{\text{Hp}} & \mathbf{S}^{\text{Hm}} \end{pmatrix}. \qquad (9)$$

For physical polarizabilities of mesh cells that compose a discretized structure, only $\mathbf{S}^{\text{Ep}} \neq 0$. It depends on the grid and dimension of the problem, for a 3d cubic discretization lattice, the self-term is:

$$\mathbf{S}^{\text{Ep}} = -\mathbf{I}\frac{4\pi}{3\varepsilon_{\text{env}} d^3} \qquad (10)$$

With the identity tensor $\mathbf{I}$. For the self-terms of the other supported discretization lattice types, see the appendix of reference[27]. It is worth noting that the here used diagonal self-terms represent only the most important correction of the Green's tensor divergence. The accuracy of the volume discretization can be further increased by including off-diagonal self-correction terms[78].

For effective point polarizabilities that describe an entire particle, the self-terms are set to zero.

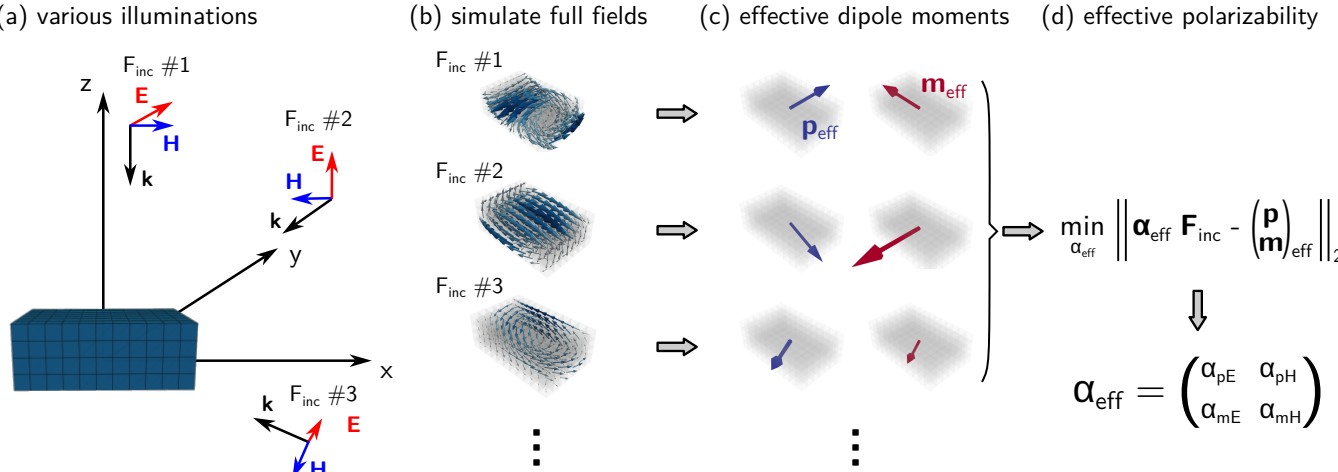

(a) various illuminations  (b) simulate full fields  (c) effective dipole moments  (d) effective polarizability

FIG. 2. Extraction of an effective polarizability model from a discretized structure. (a) define a set of illuminations. (b) simulate the full fields for each illumination. (c) for each simulation, extract the effective electric and magnetic dipole moments of the response. (d) optimize the effective polarizability to represent as closely as possible the different dipole moments.

## B. Extraction of effective models

As alternative to discretization polarizabilities, TorchGDM supports effective polarizability models, that describe the optical response of a structure in a compact model. TorchGDM implements the concept of "Global Polarizability Matrix" (GPM), similar to the idea of the T-Matrix approach, but using several spatially distributed, non-local electric and magnetic effective dipoles instead of higher order multipoles at a single expansion point. The advantage is that near-fields can be accurately described also within the circumscribing sphere. The concept is schematically depicted in figure 1b, for details we refer the reader to the original paper by Bertrand et al.[21] The most simple case of a GPM is a single effective point (3D) or line (2D) polarizability, representing an entire (sufficiently small) nanostructure. If the response cannot be truncated after the dipole order, multiple effective electric and magnetic dipole pairs can be distributed inside the structure to obtain a valid approximation. The different available structure types are illustrated in figure 1c-d.

An effective model can be extracted from a discretized structure, using the structure class methods `convert_to_gpm` and, for the specific case of a single pair of point dipoles, `convert_to_effective_polarizability_pair`. These functions take as required argument the extraction wavelengths (`wavelengths`). "GPM" extraction also requires an argument "r_gpm", which accepts either the number of effective dipole pairs to use, or a list of their positions. These functions return a new structure instance (3D-point or 2D-line effective polarizability structure), that can directly be integrated in a torchGDM simulation and combined with all other types of structures.

The effective models are generally dependent also on the angle of incidence and therefore have no exact solution. In consequence, the extraction of the effective polarizability models is an ill-posed inverse problem. We solve it in the following way: We illuminate the discretized structure with a set of $N$ different illuminations. By default, a combination of plane waves with different polarizations and incident angles and several local sources are used. The local sources are placed on random positions close to the surface (extraction from discretized nanostructures) or close to the circumscribing sphere / circle (extraction from Mie and T-Matrices). Optionally, also a list of user-defined illuminations can be used.

The procedure to extract an effective polarizability model for an arbitrary nanostructure is depicted in figure 2 by the example of a single point dipole pair polarizability model. The procedure works identically when using $N$ dipole pairs in the GPM case, except that the effective model $\boldsymbol{\alpha}_{\mathrm{eff}}$ becomes a $6N \times 6N$ matrix. In a first step we simulate the response of the fully discretized structure for all chosen illuminations. Subsequently, the effective electric and magnetic dipole moments are extracted. To this end we solve a secondary inverse problem: We optimize the electric and magnetic dipole moments such that the mismatch between scattered fields from full simulations and from the effective dipoles is minimized at a large number of probe positions. By default, again random positions close to the particle surface are used (or on a circumscribing sphere/circle when using Mie or T-Matrix). Optionally, the probe positions may also be explicitly provided by the user. This procedure is analog to what was described in related literature.[21,79] Note that in the specific case of a single effective dipole pair in 3D, we extract the dipole moments via exact multipole decomposition for the different illuminations since this provides the best accuracy.[3–5]

After this operation we have a set of $N$ illuminations $\mathbf{E}_{0,ij}(\mathbf{r}_0)$, $\mathbf{H}_{0,ij}(\mathbf{r}_0)$ and $N$ associated sets of $M$ electric and magnetic dipole moments $\mathbf{p}_{ij}$, $\mathbf{m}_{ij}$, where $i$ indicates the illumination index and $j$ the specific effective dipole. We combine these fields and moments into 2 matrices of $(N \times M, 6)$ elements each, denoted as $F_0$ for the combined electric and magnetic illumination fields, and $P$ for the combined dipole

moments. To obtain the effective polarizability model, we now need to solve the following minimization problems:

$$\min_{\boldsymbol{\alpha}_{\text{eff}}} \left\| \boldsymbol{\alpha}_{\text{eff}} F_0 - P \right\|_2 , \tag{11}$$

where $\| \ldots \|_2$ is the $L_2$ norm. We solve this linear problem using the pseudoinverse $F_0^+$ of the illumination field, which yields the optimal effective polarizability tensors to represent the full dipolar response:

$$\boldsymbol{\alpha}_{\text{eff}} = F_0^+ P , \tag{12}$$

Please note that a warning is triggered if the $L_2$ residual is larger than an empirically found threshold (which can also be manually set by the user). In this case, the structure is probably too large and different illumination directions lead to different dipole moments, which cannot be represented in the model. To solve such issues, one should increase the number of effective dipoles for the particle model. Be aware that this only tests the residuals of the minimization problem. The model may still work badly with other illuminations than used during the extraction. Therefore, the accuracy of effective models should always be tested in a small scattering simulation. This can be done automatically, by passing the parameter `test_accuracy=True` to the extraction function. This will provide a relative error with respect to cross sections and near-fields obtained from a full simulation (or a T-Matrix calculation), estimating the fidelity of the effective model approximation.

### C. Effective polarizabilities of spherical and cylindrical core-shell particles and T-Matrices

TorchGDM contains also a tool to obtain effective polarizability models for spherical and cylindrical core-shell particles as well as for arbitrary 3D and 2D T-Matrices in a homogeneous environment of real refractive index $n_{\text{env}}$. GPM models of Mie spheres, cylinders and T-Matrices are obtained using the scattered field matching technique described above. It is possible to use other Maxwell solvers to extract an effective model, either through the simulated illumination and scattered fields, or via a detour by first extracting the T-Matrix of a nanostructure[80], and convert this through TorchGDM's interface to "treams" into a global polarizability matrix effective model.

The model for a single pair of point dipoles can be obtained using the Mie scattering coefficients $a_n$ and $b_n$ (see figure 1c-d).[81,82] In the 3D case, we get:

$$\begin{aligned}
\boldsymbol{\alpha}_{pE} &= \mathbf{I} \frac{3i}{2} \frac{a_1}{k_0^3 n_{\text{env}}} \\
\boldsymbol{\alpha}_{mH} &= \mathbf{I} \frac{3i}{2} \frac{b_1}{k_0^3 n_{\text{env}}^3} ,
\end{aligned} \tag{13}$$

while in the 2D case, we have for TE polarization (electric field perpendicular to the infinite axis):

$$\begin{aligned}
\alpha_{2D,pE}^{TE} &= \frac{i}{\pi} \frac{a_1}{k_0^2} \\
\alpha_{2D,mH}^{TE} &= \frac{2i}{\pi} \frac{b_1}{k_0^2 n_{\text{env}}^2} ,
\end{aligned} \tag{14}$$

and for TM polarization (electric field along the infinite axis):

$$\begin{aligned}
\alpha_{2D,pE}^{TM} &= \frac{i}{\pi} \frac{a_0}{k_0^2} \\
\alpha_{2D,mH}^{TM} &= \frac{2i}{\pi} \frac{b_0}{k_0^2 n_{\text{env}}^2} .
\end{aligned} \tag{15}$$

$k_0 = 2\pi/\lambda_0$ is the vacuum wave number and $i$ the imaginary number.

Without limiting generality, in torchGDM the infinite axis of 2D simulations is set to be along the Cartesian $y$ axis.

It is also possible to manually define effective polarizabilities, for example using Lorentzian lineshapes.[83]

It is important to note that the Mie and T-Matrix tools are internally using the software "treams"[64], and are therefore not compatible with automatic differentiation.

### D. Hybrid discretization GDM

As mentioned before, torchGDM allows to arbitrarily combine structures approximated by effective point polarizability models and volume discretizations. In practice, we write our system of coupled equations (5) by separating all electric and all magnetic fields, as illustrated in figure 3. Technically this facilitates removal of rows and columns of zero magnetic polarizability from the system of equations, to reduce the memory footprint and computational cost for inversion.

### E. Observables

So far, torchGDM comes with implementations for the following observables:

- Near- and far-fields. Scattered fields are calculated by repropagation of the dipole moments using equation (5).

- Internal fields. Only inside discretized structures. Technically they are available only at the exact locations of the discretization cells. Fields at non-meshpoint positions are interpolated with 1/R weights using the internal fields within a 2 steps radius.

- Derived quantities from the fields, such as Poynting vector, energy flux, or near-field chirality.

- Field gradients (via automatic differentiation).

- Total extinction, absorption and scattering cross sections.[84]

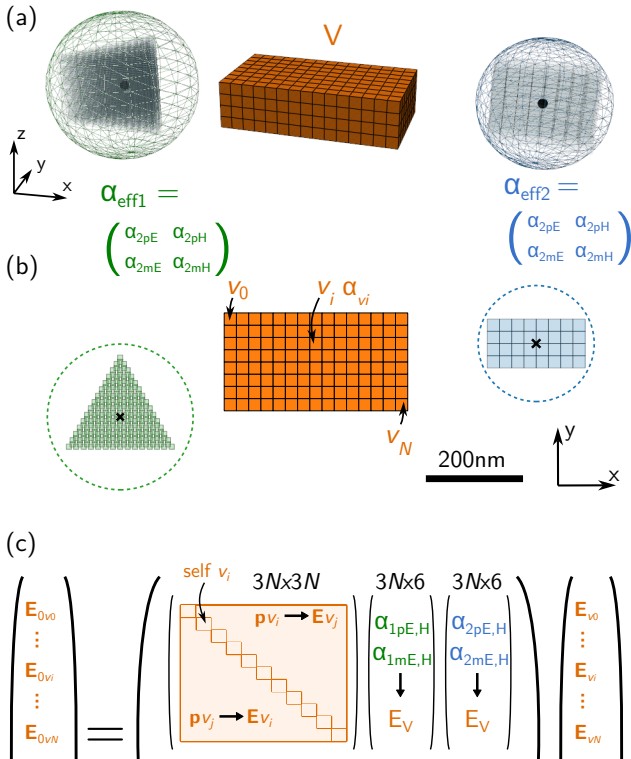

FIG. 3. (a) Example 3D geometry. A discretized structure $V$ with $N$ cubic volume elements $v_i$ (orange) is coupled to two other structures (green and blue), represented each by a pair of electric-electric and magnetic-magnetic effective polarizabilities $\boldsymbol{\alpha}_{\text{eff1}}$ and $\boldsymbol{\alpha}_{\text{eff2}}$. (b) 2D projection of the geometry. (c) Mixed polarizability linear coupled system, as solved in torchGDM (by inversion of the coupling matrix).

### F.  Implementation technical details

- The exact 3D multipole decomposition and the decomposition of the scattering cross sections are implemented as in Ref.[3] The multipole expansion of the extinction cross section is implemented following Ref.[86] Caution: Due to non-orthogonality of the modal basis for non-spherical structures, there can be cross-talk between multipoles in the extinction cross section. The exchange of energy between non-orthogonal multipole contributions can lead to the extinction coefficients for individual expansion terms becoming negative. The sum however needs to be always positive. This is discussed in more detail in works by Evlyukhin et al.[86,87]

- Effective polarizabilities at non pre-calculated wavelengths are interpolated using bi-linear interpolation. A warning is shown in these cases.

- The total scattering and absorption cross sections of GPM effective models can right now not be calculated with the optical theorem, only the extinction cross section works correctly with GPM models (models with more that 1 effective dipole pair). The effective dipoles in these GPMs are non-local: each dipole moment is a function of the illumination fields also at all other dipoles' locations. This non-local response renders the computation of the work that the field exerts on the effective dipoles complicated, and we have not implemented this so far. Currently, to obtain the scattering cross section, integration of the farfield scattering on a spherical surface is necessary. Absorption can then be obtained as $\sigma_{\text{abs}} = \sigma_{\text{ext}} - \sigma_{\text{sca,ff}}$.

- Materials permittivities from literature are supported using the "yaml" format from https://refractiveindex.info/.[88] Either tabulated data or Sellmeier models are supported so far. The permittivity data is bi-linearly interpolated between tabulated wavelengths.

- 2D Green's tensors: Due to current PyTorch limitations, Hankel functions are evaluated using a combination of Bessel functions and recurrence relations. A technical consequence is that only integer order and purely real arguments are supported so far. This can be limiting for example in applications that require integration along complex frequencies in 2D simulations.

- Using a list of many small structures in a simulation is not efficient in the current implementation. TorchGDM tries to optimize structures by combining similar types of responses before evaluation, but the used algorithm is simplistic. It is best practice to try to avoid using large lists of many small structures, i.e. combine as many effective dipoles in one structure object as possible.

- Due to memory transfer overhead, the GPU solver is less efficient for small structures (see figure 7).

- Field gradients require differentiation of many output values with respect to many input values (field components at multiple positions). Autograd is typically not efficient in such cases (neither backward mode nor forward mode). However, PyTorch offers a composable

- Total and differential scattered intensity via reprop-agation (more accurate for very small absorptive particles).[14]

- Electric and magnetic LDOS and Green's tensors in complex, nano-structured environment.[29,85]

- Exact multipole decomposition of 3D polarization distribution inside a discretized nanostructure, as well as the multipole decomposition of the scattering[3] and extinction[86] cross sections.

- Derivatives of any observable with respect to any input parameter via PyTorch's automatic differentiation interface.

functions transforms module "`torch.func`", which allows very efficient calculation of jacobians through automatic vectorization. Since `torch.func` is still in beta, torchgdm implements as a fallback alternative field gradients based on finite differences. Both methods yield similar accuracy and are automatically differentiable. Field gradient calculation may of course be also implemented using PyTorch autodiff. This will be generally slow due to the requirement of building many computational graphs, but it may be interesting in some cases, for example to evaluate higher order derivatives with high accuracy.

- By default, for a fixed wavelength all illumination field configurations are batch-evaluated in parallel by the different post-processing utilities. Especially for larger simulations this can lead to out-of-memory (OOM) errors. All concerned functions therefore support a `batch_size` argument, allowing to reduce the number of samples evaluated in parallel. If OOM errors occur, batch size needs to be reduced. TorchGDM will automatically restart failed calculations using a batch size of 1. However, this is most likely not the performance optimal configuration either. In cases of limited available memory it may therefore be worth to manually test adequate values for the batch-size.

- Most methods evaluate by default all observables (electric field, magnetic field, scattered and total fields, extinction, absorption, all multipoles, etc.). If only a single value is needed this is obviously not performance optimal. When evaluation speed is critical (for example in an optimization loop), there exist private routines for evaluation of individual observables (e.g `postproc.crosssect.ecs` or `postproc.fields._nearfield_e`). For such use-cases, see the detailed online API documentation.

### G. Automatic differentiation limitations

TorchGDM is entirely written in PyTorch, but some tools use external libraries or act as interfaces to third party software. In consequence, these functions are not compatible with automatic differentiation. We give here a list of the functionalities that are not fully torch AD-capable:

- Mie theory: All tools that use Mie theory interface with "treams"[64] and thus do not support AD. For a simulation workflow this means that Mie scattering sections are not AD-capable at all. These tools are meant to serve as a comparison or benchmark baseline. Mie-based core-shell effective models can be included in an autodiff workflow after their non-AD capable initialization.

- T-matrix conversion also uses "treams"[64] and does not support AD. T-Matrix based effective structure models can be included in an autodiff workflow after their non-AD capable initialization.

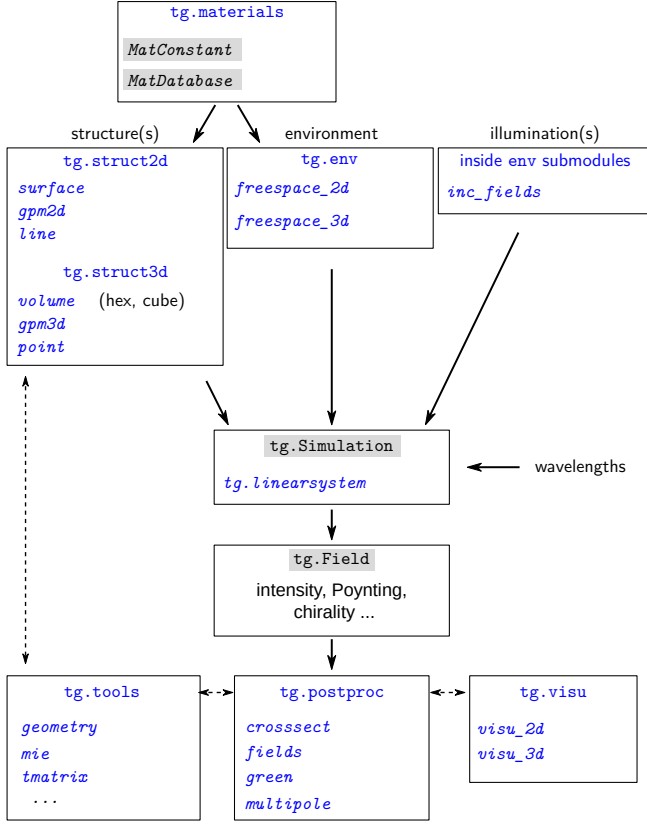

FIG. 4. TorchGDM main package structure. The entire toolkit is built around the `Simulation` class, which is a container for the structure, environment and illumination descriptions and manages the simulation. It also provides access to the most relevant post processing and visualization functions. Blue: Subpackages and modules. Black/gray: Classes.

- In the case when only an integer number of effective dipoles is given for GPM conversion, torchGDM uses spectral clustering from the "scikit-learn"[89] library to distribute effective sources inside the particle geometry. This is not AD compatible. The user can specify the effective dipole locations manually as a list of positions. In this case, GPM extraction is autodifferentiable.

- It is possible to load an image and convert it into a torchGDM discretized structure. However, the function "`tg.struct3d.from_image`" uses "PIL" to load and pre-process the image file, and therefore this operation is not autodiff capable.

### H. Package structure

We limit the presentation here to the basic package structure and the main classes. The most relevant elements of the interface are listed and discussed in the appendix A. For a full technical documentation we refer the reader to the online documentation at https://gitlab.com/wiechapeter/torchgdm.

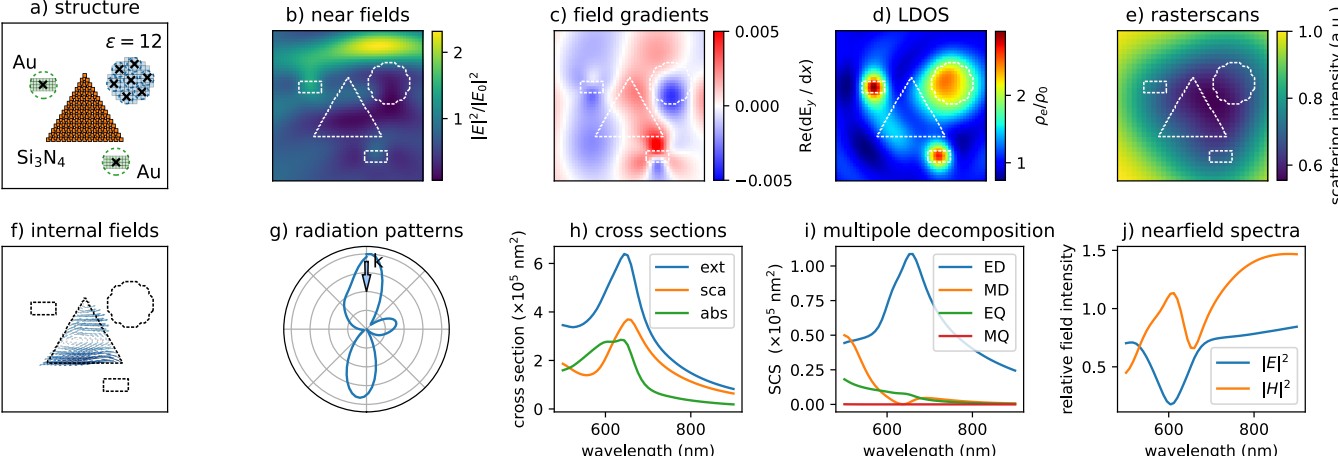

FIG. 5. Gallery of the main features of torchGDM. a) Structures can be arbitrarily composed of discretized geometries (here a dielectric prism) and effective models (here small gold nano-rods and a dielectric disc). b) Electric and magnetic near fields. c) Field gradients, either calculated via automatic differentiation or by finite differences (faster). d) Partial, electric and magnetic Local density of states as well as Green's tensors. e) Rasterscan simulations with scanning illuminations. Here, a focused Gaussian (through NA 1.4, linear $X$-polarization) is raster-scanned across the structure and the scattered far-field intensity is plotted for each beam position. f) Internal fields can be obtained for all discretized structures. g) Radiation patterns (here: cut through XY plane). h) Cross section spectra. i) Exact multipole decomposition (discretized structures only). Here, the multipole decomposition (up to quadrupoles) of the contribution of the discretized $Si_3N_4$ triangular structure to the scattering cross section is plotted. j) Near-field enhancement spectra (here 100 nm below the center of the triangle). All 2D plots show an area of $800 \times 800 \, nm^2$. Except for the LDOS and rasterscan examples, the illumination is a plane wave in the structure plane, coming from the positive $Y$ axis, with linear polarization along $X$. Single wavelength results are calculated at $\lambda_0 = 550 \, nm$. The host medium is vacuum.

*Subpackages:* TorchGDM is divided in the following sub-packages, that organize the different building blocks of a typical simulation (see also figure 4):

- `tg.env`: Defines an environment for a simulation, including the illumination light source.

- `tg.struct2d`: 2D structure classes and geometric primitives as well as a mesher for surface discretized models.

- `tg.struct3d`: 3D structure classes and geometric primitives as well as cubic and hexagonal mesher for volume discretized models.

- `tg.materials`: Provides an interface to refractiveindex.info tabulated materials.[88] TorchGDM includes a selection of database entries, but refractiveindex.info database record can be loaded from downloaded ".yaml" files.

- `tg.postproc`: Postprocessing tools to calculate physical observables from a simulation.

- `tg.tools`: Various tools for geometry manipulation, batch evaluation, interpolation, etc.

- `tg.visu`: Visualization tools (2D with matplotlib,[90] 3D with PyVista[91]).

*Main classes:* There are two main classes, defined at the top level of torchGDM. The first top-level class "`tg.Simulation`" provides an API for all simulation related postprocessing and visualization functionality of torchGDM. Complex fields (near- and far-field) are generally returned as instances of the second top-level class "`tg.Field`", which provides an API for various postprocessing and visualizations related to the electromagnetic fields.

*Results:* While electromagnetic fields are returned as instances of the `tg.Field` class, other results are usually returned in dictionaries, for which torchGDM tries to maintain a consistent nomenclature. Observables related to the scattered, total and incident field are returned in dictionary keys carrying the suffixes "sca", "tot" and "inc", respectively. Scattering, extinction and absorption cross widths (2D) and cross sections (3D) are indicated by suffixes "scs", "ecs", "acs".

*API:* Most functions are accessible through an object oriented interface from the torchGDM classes, but also through a functional interface in the different subpackages. For a detailed discussion of all submodules, functions and class methods, please see Appendix A or the online API documentation (https://gitlab.com/wiechapeter/torchgdm).

## III. EXAMPLES

In the following, we demonstrate the key features of torchGDM by a choice of a few examples. Their full codes,

```
1  # ------ minimum full example
2  import torch
3  import matplotlib.pyplot as plt
4  import torchgdm as tg
5
6  # - vacuum environment
7  env = tg.env.EnvHomogeneous3D(env_material=1.0)
8
9  # - illumination field(s)
10 plane_wave = tg.env.freespace_3d.PlaneWave(
11   e0p=1.0, e0s=0.0
12 )
13
14 # - discretized structure
15 structure = tg.struct3d.StructDiscretizedCubic3D(
16   discretization_config=tg.struct3d.volume.cube(l=8),
17   step=30,
18   materials=tg.materials.MatDatabase("GaN"),
19 )
20
21 # - define and run simulation
22 sim = tg.Simulation(
23   structures=[structure],
24   illumination_fields=[plane_wave],
25   environment=env,
26   wavelengths=torch.linspace(550.0, 950.0, 100),
27 )
28 sim.plot_structure()
29
30 # run the simulation
31 sim.run()
32
33 # calculate cross sections
34 cs_results = sim.get_spectra_crosssections()
35
36 # plot
37 plt.plot(
38   tg.to_np(cs_results["wavelengths"]),
39   tg.to_np(cs_results["ecs"]),
40 )
41 plt.show()
```

Listing 1. Minimum example script. The plots generated by the script (structure and spectrum) are shown in Fig. 6.

as well as further examples can be found in the online documentation.

### A. Capabilities

To give an overview of what can be done with torchGDM, some of the core capabilities are summarized in figure 5. An extensive list of all implemented observables is given in the Appendix or in the online documentation. All shown calculations are fully compatible with PyTorch automatic differentiation.

So far, only homogeneous media are supported, but torchGDM can be easily extended without modification of the core code through custom environment classes. An example showing how to implement a host medium with a dielectric interface through a mirror dipole approximation is given in the online documentation.

### B. Minimum Example

Listing 1 showcases a code example for a simple scattering simulation. It sets up a 3D vacuum environment and a linear

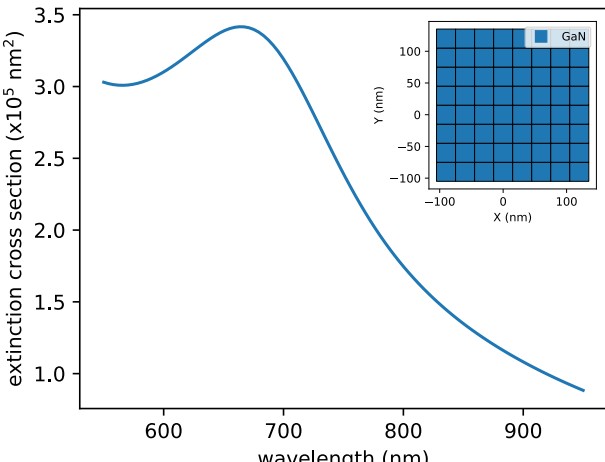

FIG. 6.    Spectrum obtained by the minimum example in listing 1.    The inset shows the structure plot, generated by `sim.plot_structure()`.

polarization plane wave illumination. As structure, a dielectric nano-cube is created, made from Gallium Nitride, with a side length of $8 \times 30$ nm steps. This is wrapped into a simulation, that is evaluated at 100 distinct wavelengths between $\lambda_0 = 550$ nm and $\lambda_0 = 950$ nm. After running the simulation, the extinction cross section spectrum is calculated and plotted, shown in figure 6.

### C. Benchmarks

*Runtime and memory consumption*

Figure 7a shows a benchmark of runtime vs number of simulated dipoles for evaluation on a single CPU core (blue), using 16 CPU cores (orange) and using a CUDA GPU (green). Note that by default all available CPU cores are used, the single core example is for illustration of the parallelization efficiency, which lies somewhere between 0.5 and 0.8, depending on the system. While communication overhead slows down CUDA for small simulations, from a few 100 dipoles on, running simulations on GPU is by far the most efficient choice.

Figure 7b shows the memory requirement, which is identical for whichever hardware is used. Because so far torchGDM only implements a full solver, the memory requirement increases with $N^2$ ($N$ being the number of dipoles). This limits the possible size to around 10,000-15,000 dipoles. In the future we might implement an iterative solver to enable larger simulations.

Figure 8 shows the performance impact if using automatic differentiation (on CPU). The required compute (Fig. 8a) approximately doubles due to the backward pass calculation, but it does not depend significantly on the type and number of derivatives. The required memory (Fig. 8b) increases also, and more memory is necessary the more partial derivatives are to be calculated.

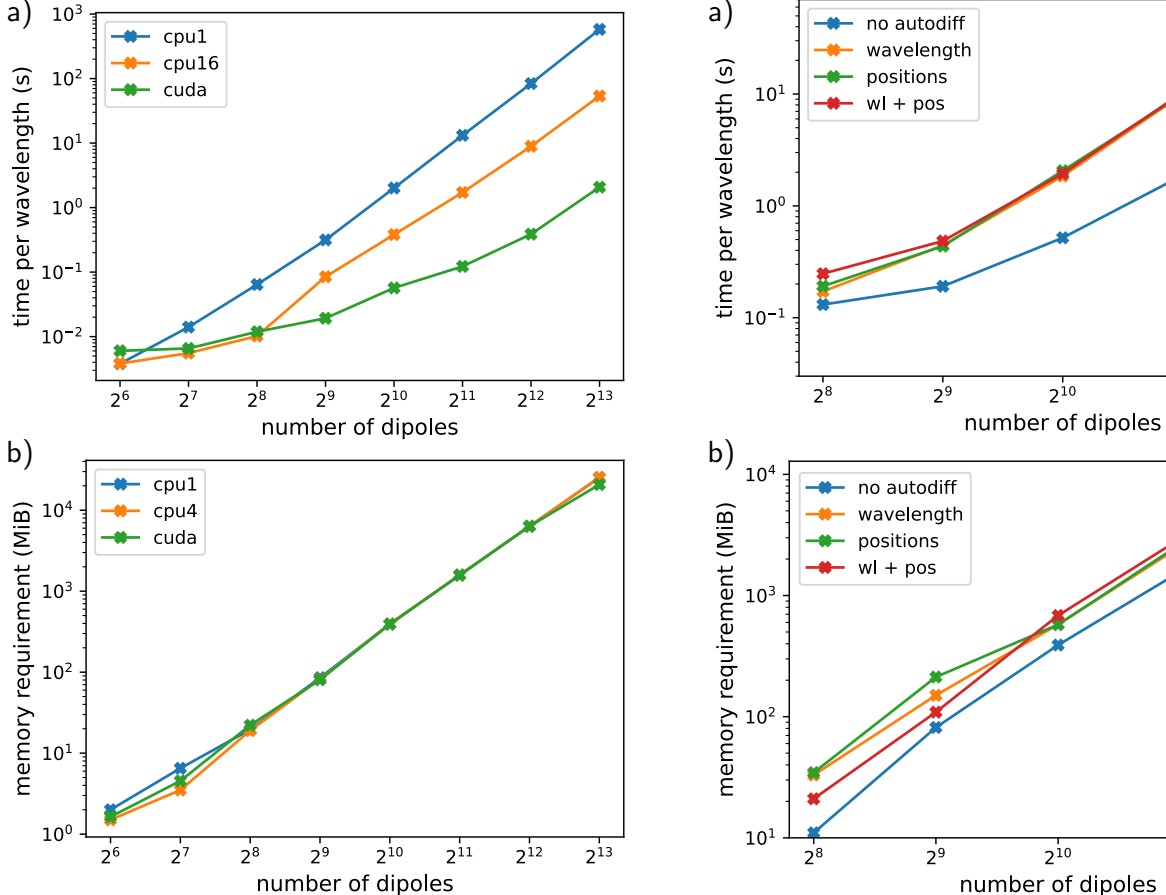

FIG. 7. Benchmark without automatic differentiation. a) TorchGDM calculation time per wavelength on a single CPU core (blue), on 16 CPU cores (orange) as well as using a CUDA GPU (green). b) memory requirement (CPU: RAM, GPU: VRAM) for the same configurations. CPU is an AMD Threadripper 3970X, GPU is an NVIDIA RTX 4090.

FIG. 8. Benchmark with automatic differentiation. a) TorchGDM simulation runtime per wavelength without automatic differentiation (blue) and with automatic differentiation with respect to the wavelength (orange), the structure dipole positions (green) and both (red). b) memory requirement for the same configurations. The benchmarks are done on 8 CPU cores. Trends are similar for GPU timing and memory requirements. The same system was used as for the benchmarks in figure 7.

### *Accuracy of effective dipole models*

Figure 9 demonstrates the accuracy of the global polarizability matrix effective models for a complicated 2D scatterer, using a difficult illumination. The structure is shown in figure 9a, it is made of a lossless dielectric with $\varepsilon = 9$, and is placed in vacuum. The discretized model consists of a bit more than 3700 dipoles, the equivalent effective model is made from 20 dipole pairs, corresponding to a reduction of roughly $\times 100$. The illumination is a line dipole along $Y$, placed within a notch of the geometry. The location of the local light source is well within the circumscribing circle of the particle, thus this situation cannot be modelled with conventional T-Matrix in vector spherical harmonics expansion. Figure 9b shows an excellent agreement between the farfield scattering patterns of the discretized model vs the global polarizability matrix. Figure 9c-d compare the electric nearfield intensity maps of the two cases. Outside of the extraction probe and source locations (green and red dots in Fig. 9a),

the near-fields are very accurately modelled by the GPM (also within the circumscribing circle).

### *Comparison with Mie theory*

Figure 10 demonstrates the accuracy for a single scatterer by a comparison of full discretization and Mie-based effective polarizability models with analytical Mie theory. Figure 10a shows the case of a 3D GaN sphere, Fig. 10b shows results for an infinite GaN cylinder (2D), both placed in vacuum. Simulations using effective polarizability models are matching Mie theory, the simple dipole-only models of course are valid only until higher order modes occur. The discretized simulations have the highest error, which is due to imperfect spherical approximation of the geometry with the coarse discretization.

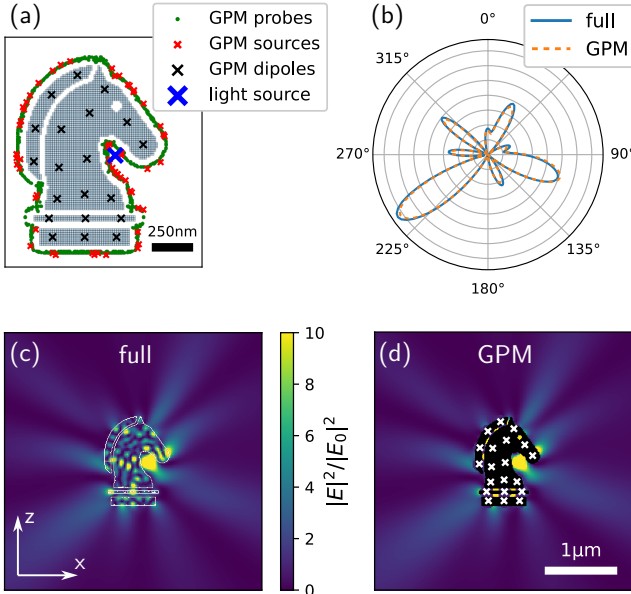

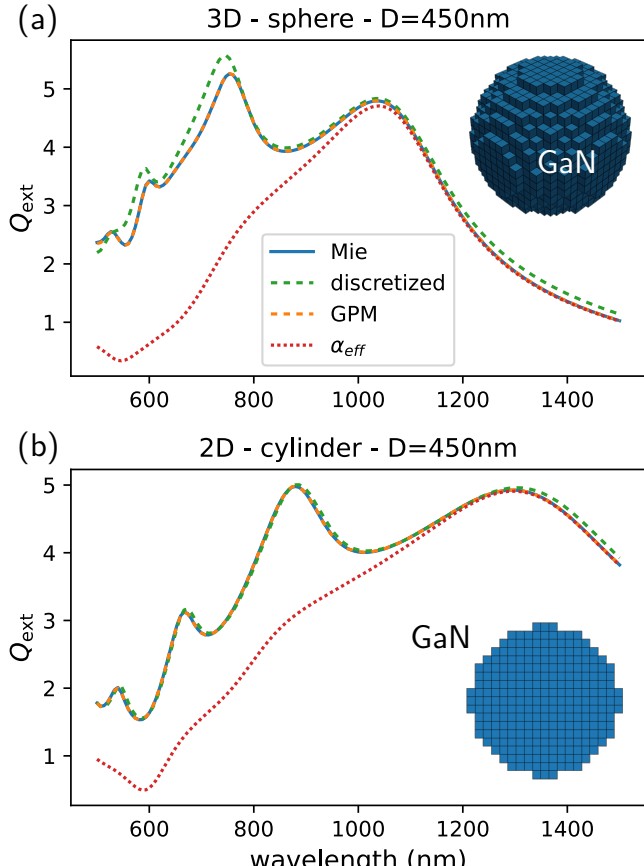

FIG. 9. Demonstration of the accuracy of the effective models in comparison with the full discretization of a complex 2D nanostructure. (a) Sketch of the geometry and the effective model (black markers). Red and green markers indicate, respectively, the source and probe locations used for the effective model extraction. The full discretization is composed of roughly 3700 dipoles on a 2D square mesh with a step of 14.2 nm. The effective GPM structure is a global polarizability matrix (GPM) based on 20 dipole pairs. The structure is placed in vacuum and is made of a lossless dielectric with $\varepsilon = 9$. It is illuminated by a line dipole oscillating along the $Y$ axis, placed at the location indicated by the blue marker. (b) Radiation pattern of full discretization (solid blue line) vs effective dipole model (dashed orange line). (c) Total electric nearfield intensity calculated from the discretized structure. (d) Same as (c) but calculated using the effective model. White markers indicate the GPM effective dipole locations.

*Comparison with the T-matrix method*

The coupled effective dipole description is formally identical to the T-matrix method at dipolar order. Therefore, using small enough spheres described by polarizabilities derived from analytical Mie theory, torchGDM results exactly match Mie theory. On the other hand, GPMs with multiple dipoles per structure are empirical models, obtained by matching the scattered fields outside of the particle and therefore will not exactly match Mie theory or T-Matrix calculations. However, in practice the errors are generally very low in the order of $10^{-3}$ or less. Keep in mind that when using volume discretization for polarizability model extraction, the accuracy cannot exceed the fidelity of the discretized simulation. The additional error depends primarily on the number of dipoles in an effective model. Using 10 to 15 dipole pairs is a good rule of thumb to get a very good effective model even for larger structures. A comparison to the T-matrix method is shown in figure 11, further examples comparing 2D and 3D geometries with T-matrix calculations as well as a demonstration how effective models can be obtained from T-matrices are available

FIG. 10. Polarizability extraction accuracy: Mie theory (solid blue lines) is compared to discretization (green, dashed), global polarizability matrix (GPM, orange dashed) and a dipole-pair model (red, dotted). (a) 3D case of a sphere and (b) 2D case of an infinite cylinder. The diameters of the sphere and of the cylinder are both $D = 450$ nm. The embedding environment is vacuum, and the material is Gallium Nitride (GaN) in both cases. (a) Cubic discretization with step of 25 nm. (b) Square discretization with step 15 nm. The illumination is a linearly polarized plane wave. In (b) polarization is along the cylinder axis (transverse magnetic, TM). The effective polarizability models are obtained from Mie theory. The 3D GPM model uses 15 dipole pairs, the 2D GPM uses 5 dipole pairs. The single effective dipole approximation ("$\alpha_{\text{eff}}$") is valid only in the long wavelength region.

in the online documentation.

### D. Example: Bound States in the Continuum

Arrays of effective structures can be used to simulate lattice effects in limited size periodic structures, such as bound states in the continuum (BICs), which are inaccessible from the far-field. Through symmetry breaking these states (then "quasi BICs") can become accessible. We demonstrate the possibility to simulate this by reproducing recently reported results from Dong et al.,[92] where small assymetries are introduced by an opposite tilting angle of neighbor structures within a unit

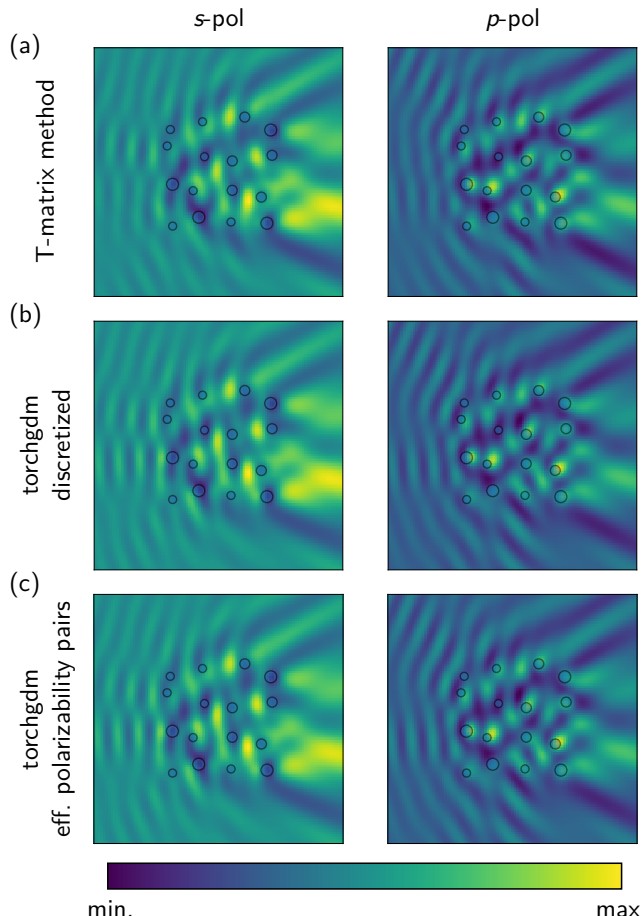

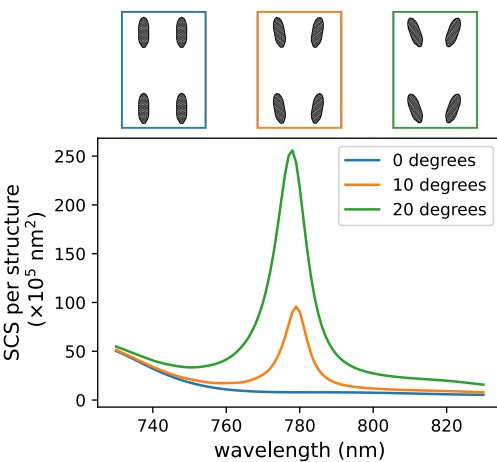

FIG. 12. Normal incidence plane wave illumination. Unit cells are shown in the insets, the full simulated structure consists of $15 \times 15$ unit cells (in total 900 elliptical structures).

```
1  # ------ automatic differentiation usage
2  # - set require gradients
3  wavelengths = torch.as_tensor([700])
4  wavelengths.requires_grad = True
5
6  # - define and run simulation
7  sim = tg.Simulation(
8    ...,
9    wavelengths=wavelengths
10 )
11 sim.run()
12
13 # - calculate *anything*, e.g. the ecs:
14 res_ext = tg.postproc.crosssect.ecs(
15    sim, wavelength=wavelengths[0]
16 )
17 ecs = res_ext["ecs"]
18
19 # - gradient: d ecs / d wavelength:
20 ecs.backward()
21 decs_dwl = wavelengths.grad[0]
```

Listing 2. Autograd example code. This code is used for the gradient based search of the closest resonance in Fig. 13.

FIG. 11. Multi-scattering accuracy: Comparison with T-matrix method. A random arrangement of dielectric spheres ($\varepsilon_{sph} = 9$) in vacuum ($\varepsilon_{env} = 1$) with radii of 80 nm, 100 nm and 120 nm is laying in the $XY$ plane. The ensemble is illuminated from the left by a plane wave with $\lambda_0 = 800$ nm wavelength. The electric field intensity is calculated at $z_0 = -250$ nm below the plane of the sphere centers, using different methods. (a) T-matrix method using "treams", (b) fully discretized (cubic, step of 20 nm) and (c) using effective ED, MD dipole pairs. Shown areas are $5 \times 5 \, \mu m^2$. The colormaps are normalized to the same scale in all panels.

### E. Example: Autograd for resonance search

The key capability of torchGDM is automatic differentiation through *any* simulation. Technically, this is simply using pytorch's autodiff (AD) mechanism, and therefore works exactly as any other application of pytorch's autograd.

PyTorch's AD can be applied to any continuous variable that occurs in any calculation. As a simple example, we demonstrate in listing 2 how to use automatic differentiation on the illumination wavelength to find resonances of a photonic nanostructure (see figure 13).

### F. Example: Huygens metalens optimization

The multiscale capabilities of TorchGDM together with automatic differentiation are interesting for optimization applications of photonic structures with complicated response

cell of the lattice. This symmetry breaking allows coupling of a far-field source to the quasi-BIC, as shown in figure 12.

While it is common practice to use the coupled dipole method for simulation of (quasi-)BICs,[93,94] torchGDM offers two particularities: (1) the possibility to use automatic differentiation, which is interesting e.g. for stability tests, illumination engineering or design optimization. (2) the possibility to fully discretize some of the elements, which enables to perform simulations with local sources like quantum emitters or fast electron beams, in very close proximity or even within a single structure, e.g. in a quasi-BIC supporting lattice.

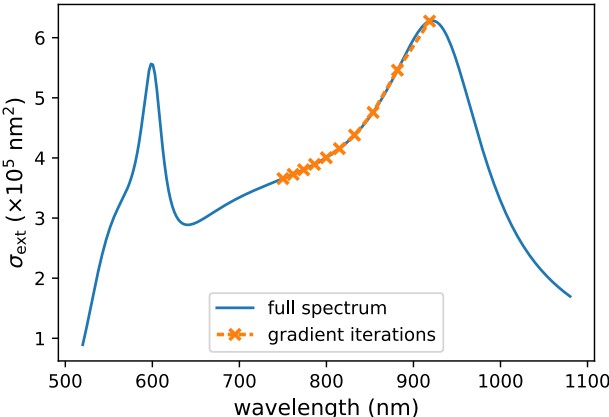

FIG. 13. Automatic differentiation for resonance search: The gradient of the extinction cross section with respect to the wavelength is iteratively calculated via AD to find the wavelength of maximum $\sigma_{ext}$ (orange markers). The full spectrum is shown for comparison (blue).

```
1  # ------ hybrid mesh / eff. dipole simulation
2  # - setup
3  env = tg.env.freespace_3d.EnvHomogeneous3D(env_material
       =1.0)
4  wl = 600.0
5  e_inc = tg.env.freespace_3d.PlaneWave()
6
7  # - discretized structure
8  mat = tg.materials.MatConstant(eps=6.0)
9  struct_mesh = tg.struct3d.StructDiscretizedCubic3D(
10   tg.struct3d.volume.cube(8), step=15, materials=mat
11 )
12
13 # - eff. pola. structure
14 struct_eff = struct_mesh.convert_to_gpm(
15   r_gpm=5, wavelengths=wl, environment=env
16 )
17 struct_eff += [150, 450, 0]   # move structure
18
19 # - create a mixed simulation
20 sim_mixed = tg.simulation.Simulation(
21   structures=[struct_mesh, struct_eff],
22   environment=env,
23   illumination_fields=e_inc,
24   wavelengths=wl,
25 )
26
27 sim_mixed.run()
```

Listing 3. Simple mixed discretization simulation setup example.

and/or long-range optical interactions. We demonstrate the potential of automatic differentiation for design tasks in figure 14, which shows how a diffractive lens, composed of dielectric cuboids, is created, by applying AD on the field intensity at the target focus position with respect to the positions of all the structures. The structures are initially on a perfect lattice. The optimization converges to a concentric structure which efficiently focuses the incoming plane wave.

A related application, where this approach seems extremely promising are Huygens metasurfaces, composed of dipolar resonant nanostructures. They are very difficult to design, conventional lookup tables typically don't work, because of the strong dependence of the electric and magnetic dipole modes to the local field of varying neighbors.[95] TorchGDM can use automatic differentiation for an inverse design based on a holistic description of the full structure with all optical interactions taken into account.

### G.    Example: Mixed discretization. Quantum emitter within a hollow nanostructure close to many scatterers

As said before, next to automatic differentiation, the second key particularity of torchGDM is the possibility to combine effective polarizability models and fully discretized structures in the same simulation. Listing 3 below shows how to define such a simulation.

This method allows for example, to place a local emitter like a fluorescent molecule or a quantum dot, very close to a nanostructure of complex shape, which itself lies within a large, macroscopic assembly of many nanostructures, e.g. a periodic array or some other arrangement. The nanostructure(s) that is(are) closest to the local light source, can be fully discretized, while the structures farther from the source can be excellently approximated each with an effective polarizability model.

Figure 15 demonstrates this scenario by the example of a local emitter coupled to a plasmonic split-ring resonator, which is in the center of an optical corral composed of resonant dielectric rods.[96,97] Such a system is typically difficult to describe because of the different involved scales: The emitter's emission and the plasmonic response require fine discretization, but also the contributions of the optical Corral are essential for the global optical response (Fig. 15a). The T-Matrix method cannot be used because the emitter lies within the circumscribing sphere of the split ring structure. A global polarizability matrix can in principle be used to describe the split-ring antenna with the local light source (Fig. 15c,e), however, optimizing the extraction process such that the emitter field within the gap is correctly modeled would require manual optimization of the GPM extraction procedure. TorchGDM's automatic GPM fit gives a model that offers only a qualitative agreement (Fig. 15c,e). A mixed discretization however, reproduces the full simulation with errors of the order of $\pm 1\%$ (Fig. 15b,d).

### H.    Example: Full fields of large structure assemblies via iterative hybrid discretization evaluation

As a final example, we demonstrate in figure 16 how mixed effective polarizability / discretization simulations can be used to recover full fields from very large simulations. We consider a large assembly of $N$ complex shape particles (here L-shaped silicon corners), that have by themselves a complex optical response including higher order contributions like quadrupoles. We extract an effective GPM model for the L-shape, consisting in six dipole pairs, which approximates accurately the optical response. However, the drawback of the GPM model is, that internal fields cannot be accessed. But we can use

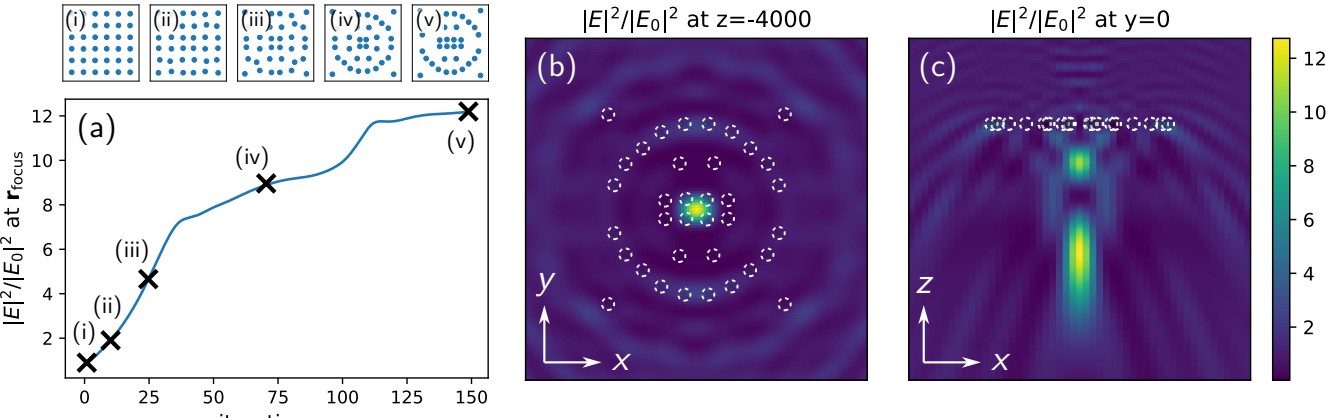

FIG. 14. Example of a diffractive lens design by gradient optimization of the positions of silicon cuboids of size $250 \times 250 \times 125 \, \text{nm}^3$ in the $xy$ plane. (a) Improvement of the field intensity enhancement at the target focus position (0,0, -4000 nm) over 150 iterations of gradient optimization. The insets show the geometries at selected moments during the optimization, indicated by cross markers with the same labels. Inset areas are $6.4 \times 6.4 \, \mu\text{m}^2$. (b) field intensity map of the final design in the focal plane (parallel to $XY$ at $z = -4000 \, \text{nm}$). (c) field intensity map in the $XZ$ plane (parallel to the illumination propagation direction). Both maps show areas of $9 \times 9 \, \mu\text{m}^2$. Illumination is an $x$-polarized normal incident plane wave from the positive towards the negative z-axis, in vacuum, wavelength $\lambda_0 = 775 \, \text{nm}$.

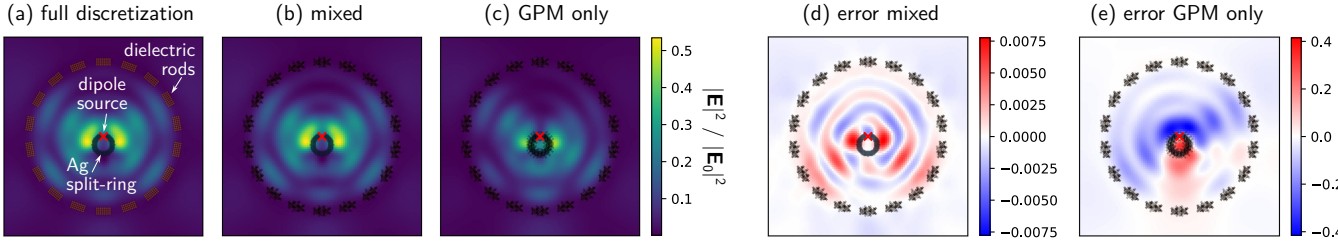

FIG. 15. Example of a "mixed discretization" simulation. An optical corral composed of resonant dielectric nano-rods ($\varepsilon = 10$) contains a plasmonic split-ring resonator (made from silver). The system is in vacuum, illuminated by an electric dipole emitter (red cross) in the gap of the split ring with $\lambda_0 = 500 \, \text{nm}$. (a-c) Total electric field intensity maps 100 nm below the structure surface. Panel (a) shows the fully discretized simulation. Replacing the dielectric rods by global polarizability matrices and keeping the split-ring in full discretization (b) gives an excellent approximation to the full simulation, with errors well below $\pm 1\%$ (d). On the other hand, replacing also the split-ring by an effective GPM model (here: 20 dipole pairs), merely gives a qualitative approximation (c) with errors up to 40% (e).

TorchGDM's capability to combine different structure types in a single simulation, in order to recover the full fields. To do so, we run a simulation where the closest neighbors around a considered structure are fully discretized, to obtain the best possible accuracy for close neighbor interactions. Structures farther away than a specific distance (here we arbitrarily chose $1.25 \times \lambda_0$), are described by an effective polarizability model (see Fig. 16a). Running $N$ simulations, one "centered" on each nano-structure, we finally recover a very accurate approximation ($\lesssim 0.5\%$ relative error, see Fig. 16b) for the full internal fields of all nanostructures. Those full fields can furthermore be used to calculate the scattered fields of the full ensemble at a higher accuracy than with the GPM approximation alone (see figures 16b-g). This technique allows to recover full fields of otherwise prohibitively large geometries.

## IV. CONCLUSIONS

In conclusion, in this work we presented "TorchGDM", a python toolkit for 2D and 3D electrodynamic scattering simulations, with focus on nano-optics applications. The toolkit has two main particularities: (1) TorchGDM supports mixed simulations combining volume discretized structures with Global Polarizability Matrix models (multiple, non-local electric/magnetic effective dipoles). These "GPMs" are similar to the T-Matrix approach, but can model fields also inside the circumscribing sphere around a particle. We believe that this will be very useful for multi-scale simulations. (2) TorchGDM is fully written in PyTorch, a modern automatic differentiation framework. This makes any calculation differentiable. We forsee that this has great implications for various applications such as design optimization, sensing or pulse-shaping. It also means that the toolkit is directly compatible with deep learning models, opening various perspectives in this context, such as physics informed learning. Finally, every

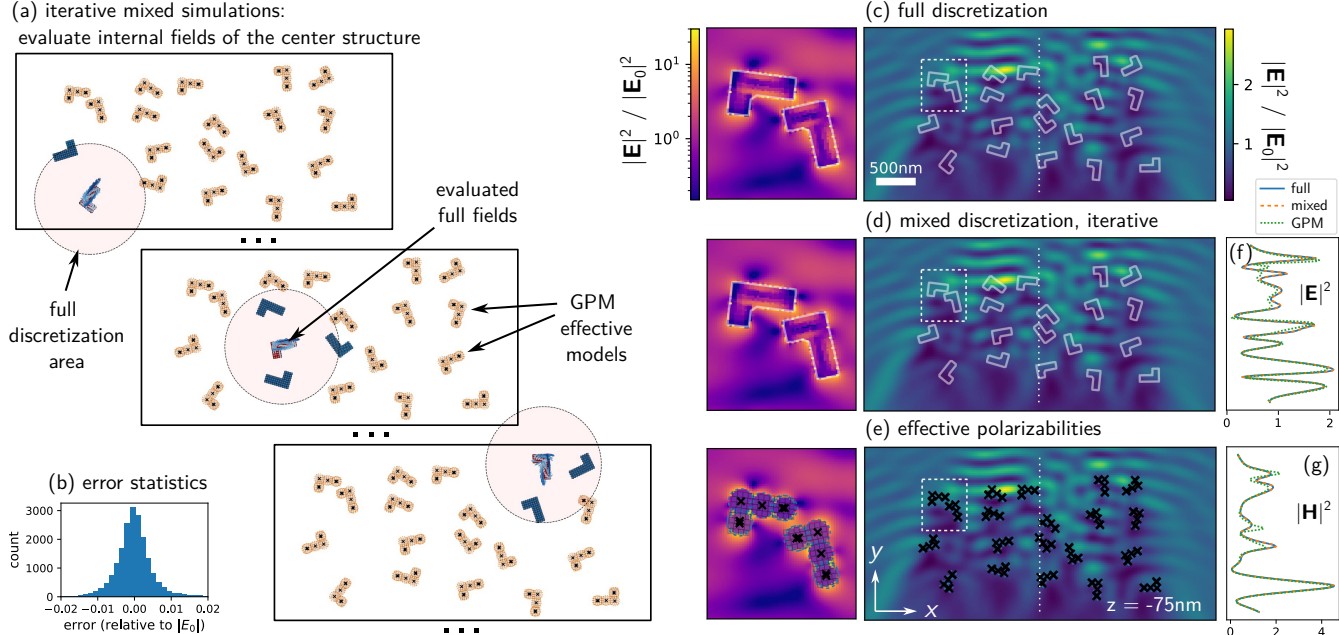

FIG. 16. Example of full field evaluation in large, complex structure assemblies with iterative, mixed discretization simulations. (a) Sketch of the procedure, "focusing" on each structure separately and performing a mixed discretization simulation. One simulation per structure is performed, the full fields are eventually combined. The geometry in this example consists of 20 L-shaped silicon structures, each with arm lengths of 300 nm (long arm) and 200 nm (short arm). The width and height of the structure are 100 nm and 125 nm, respectively. The structures are positioned on a randomly perturbed grid, with random rotation angles. The GPM model consists of 6 effective dipole pairs. (b) statistics of the error of the reconstructed full fields, the standard deviation is below 0.5%. (c-e) Field intensity maps at 75 nm below the structure bottom surface. Calculated (c) with a single, fully discretized simulation. (d) with the iterative, mixed discretization procedure. (e) with all structures modelled as GPMs. The zoom insets on the right show an $700 \times 800\,\mathrm{nm}^2$ large area, cut through the center plane of the nano-structures on a logarithmic colorscale, demonstrating the faithful reconstruction of near- and internal fields. Nearfields within the circumscribing sphere are also well reproduced with only GPMs. Illumination is an $x$-polarized plane wave incident from the positive towards the negative y-axis in vacuum, wavelength $\lambda_0 = 550\,\mathrm{nm}$. (f-g) show electric and magnetic field intensity profiles very close (25 nm) above the structure surface (evaluated along the dotted vertical line indicated in the field maps).

part of the simulation is represented by a python class, making it highly extensible. In addition to the here shown examples, the online documentation features various tutorials that demonstrate how to implement custom extensions (such as materials, structures, illuminations, or environments), alongside with a large selection of additional examples. The toolkit is freely available under an open source licence.

## AVAILABILITY AND SOURCE CODE

The source code of torchGDM is available at the dedicated git repository under https://gitlab.com/wiechapeter/torchgdm. It is also available on the PyPi repository https://pypi.org/project/torchgdm/ and can be conveniently installed via "pip" (`pip install torchgdm`). It was tested with python version 3.9 to 3.12 on linux, windows and Mac.

## ONLINE DOCUMENTATION

Extensive online documentation can be found at https://homepages.laas.fr/pwiecha/torchgdm_doc/.

## ACKNOWLEDGMENTS

We thank Otto L. Muskens, Kevin Vynck, Antoine Monmayrant, Antoine Rouxel, and Dalin Soun for fruitful discussions and for testing. This work was supported by the French Agence Nationale de la Recherche (ANR) under grants ANR-22-CE24-0002 (project NAINOS) and ANR-23-CE09-0011 (project AIM), and by the Toulouse HPC CALMIP (grant p20010).

## APPENDIX

### A. API DETAILS

We present in this appendix the most relevant elements of the torchGDM interface. For a full technical documentation we refer the reader to the online documentation.

## 1. Main classes and their methods

There are two main classes, defined at the top level of torchGDM. The first top-level class " `tg.Simulation` " provides an API for all simulation related postprocessing and visualization functionality of torchGDM. Complex fields (near- and far-field) are generally returned as instances of the second top-level class " `tg.Field` ", which provides an API for various post-processing and visualizations related to the electromagnetic fields. All those functions are also accessible through a functional interface. Here we limit the description to the object oriented API.

### The `tg.Simulation` class

The `Simulation` class is first of all a container for all parts that describe the simulation. As depicted in figure 4, these ingredients are passed to the `Simulation` upon creation, and are:

- `structures`: a list of structures (which by themselves contain their materials or polarizabilities).

- `environment`: the simulation environment (from `tg.env`).

- `illumination_fields`: a list of illumination fields (available inside the respective environment's submodule).

- `wavelengths`: the evaluation wavelengths (in nm).

The `Simulation` class manages the solver. The solver class is defined in the module `tg.linearsystem`, and is called in the simulation through the method

- `run()` Runs the scattering simulation

No manual configuration is required as currently only a full inversion is implemented, limiting the size of the simulation to around 10000 dipoles. Furthermore, several convenience operations are provided by the `Simulation` class

- `copy()` Returns a copy of the simulation, optionally with all structures shifted by a vector.

- `add_struct()` Adds a structure to the simulation.

- `delete_struct()` Deletes a structure from the simulation.

- `combine()` Combines several simulations together, conserving any pre-calculated fields. Wavelengths and illuminations of both simulations must match.

- `split()` Splits off one of the structures into a separate simulation object.

Finally, the `Simulation` class provides an interface for accessing the simulation results as well as for post-processing and visualization. There are two generic methods, that need to be provided with an evaluation function from the postprocessing subpackage `tg.postproc`:

- `get_spectra(func)`: Generic spectrum calculation for a given evaluation function "func", calculated at all available wavelengths.

- `get_rasterscan(func)`: (for position-varying illuminations like a focused Gaussian or point sources) Calculates a rasterscan for a given observable at a specific wavelength, by evaluating all illumination positions.

Further post processing methods can be categorized into "single wavelength" and "spectrum" evaluation. Following methods evaluate observables at a single, specified wavelength:

- `get_crosssections()`: Returns cross sections at a specific wavelength (extinction, scattering and absorption).

- `get_farfield()` or `get_ff()`: Returns electric and magnetic fields for given probe locations in the far-field region.

- `get_nearfield()` or `get_nf()`: Returns electric and magnetic fields (total, scattered and incident) for given probe locations in the near-field region. Outside of the source zone via repropagation. Inside of structures, via interpolation from the fields at the meshpoint locations.

- `get_nearfield_intensity_efield()`: Computes the electric field intensity at given positions in the nearfield region.

- `get_nearfield_intensity_hfield()`: Computes the magnetic field intensity at given positions in the nearfield region.

- `get_chirality()`: Calculates the nearfield chirality at a given wavelength and positions.

- `get_poynting()`: Calculates the Poynting vector for total, scattered and incident fields at a given wavelength and positions.

- `get_energy_flux()`: Returns the time averaged Poynting vector for total, scattered and incident fields at a given wavelength and positions.

- `get_field_gradients()`: Returns gradients of the nearfields inside or in proximity of the modeled structure at a given wavelength and positions.

- `get_green()`: Computes the Green's tensor of the complex environment defined by the modeled structures.

- `get_ldos()`: Computes the electric and magnetic LDOS of the complex environment defined by the modeled structures.

- `get_geometric_crossection()`: Returns a set of geometric cross sections for all modelled structures (in $nm^2$).

Spectral evaluation of specific observables (at all wavelengths defined in the simulation):

- `get_spectra_crosssections()`: Returns cross section spectra (extinction, scattering and absorption).

- `get_spectra_nf()`: Returns spectra of complex nearfields at given positions. A `Field` instance is returned for each wavelength.

- `get_spectra_nf_intensity_e()`: Computes spectra of electric field intensities in the nearfield region, integrated over probe points.

- `get_spectra_nf_intensity_h()`: Computes spectra of magnetic field intensities in the nearfield region, integrated over probe points.

- `get_spectra_ff_intensity()`: Computes spectra of integrated total, scattered and incident far-field intensities.

- `get_spectra_chirality()`: Calculates spectra of the field chirality.

- `get_spectra_multipole()`: Calculates the spectra for the exact multipole moments to quadrupole order. Optionally the long-wavelength approximation can be used.[S3]

- `get_spectra_multipole_scs()`: Calculates the scattering cross section spectra for the exact multipole moments till quadrupole order.

- `get_spectra_multipole_ecs()`: Calculates the extinction cross section spectra for the exact multipole moments till quadrupole order.

- `get_spectra_ldos()`: Computes the LDOS spectra.

- `get_spectra_green()`: Computes spectra of the Green's tensor for the full simulation system.

The `Simulation` class also contains methods for visualization of the structure or internal fields:

- `plot_structure()`: Plots 2D projection of the structure(s).

- `plot_contour()`: Plots a 2D projection of the contour(s) of the structure(s).

- `plot_structure_3D()`: Plots the structure(s) in 3D.

- `plot_efield_vectors_inside()`: 2D projected Quiver plot of the electric field inside.

- `plot_hfield_vectors_inside()`: 2D projected Quiver plot of the magnetic field inside.

- `plot_efield_vectors_inside_3D()`: 3D Quiver plot of the electric field inside.

- `plot_hfield_vectors_inside_3D()`: 3D Quiver plot of the magnetic field inside.

### Classes in `tg.materials`

Contains the class `MatConstant` for fixed permittivity materials and the class `MatDatabase` for tabulated or Sellmeier permittivity models, compatible with yaml files from refractiveindex.info.[S88] `MatDatabase` can either be called using the name of one of the materials available in TorchGDM (can be obtained using `tg.materials.list_available_materials()`), or by specifying the path name pointing to a yaml file from a refractiveindex.info full database entry (`MatDatabase(yaml_file=path_to_file)`).

The former two classes implement isotropic materials. An example showing how anisotropic permittivities can be implemented is given by the class `MatTiO2` for the birefringent permittivity of crystalline $TiO_2$ (see the online documentation).

Available methods:

- `get_epsilon(wavelength)`: Returns the complex permittivity tensor at "wavelength" (in nm).

- `plot_epsilon()`: Plots the permittivity spectrum.

- `plot_refractive_index()`: Plots the spectrum of the refractive index $n = \sqrt{\varepsilon}$.

### Classes in `tg.env`

Currently, two environment are available: a homogeneous 3D environment is implemented through the class `tg.env.EnvHomogeneous3D`. A homogeneous 2D environment is available with the class `tg.env.EnvHomogeneous2D` (assuming an infinite axis along $y$). Furthermore, each environment comes with its own set of illumination fields (see below). The so far implemented environment only support an isotropic, lossless host medium. As mentioned above, all simulations can contain a mix of volume discretized particles and effective point-polarizabilities.

All environment classes provide various methods for Green's tensor evaluation. For a detailed documentation we refer the interested reader to the online documentation, which describes every available method. The most relevant method is the full Green's tensor retrieval:

- `get_G_6x6()`: Returns full electric magnetic dipole Green's tensor. Internally, to fill the full tensor, this calls the above four separate methods.

**Illumination classes**

TorchGDM contains various illumination field definitions. The illumination classes are defined in a module `inc_fields` inside the corresponding environment subpackage. All illumination classes provide a method for field evaluation:

- `get_field(r, wavelength, environment)`: Returns a `Field` instance with the electric and magnetic fields.

All illumination classes implement further methods for separate E and H-field evaluation (`get_efield` and `get_hfield`), among others. A complete description can be found in the online documentation.

*2D Illuminations in `tg.env.freespace_2D.inc_field`*

- `PlaneWave` : A plane wave. Supports any kind of polarization and oblique incident angles in the XZ and YZ planes. Note: if the parallel wave vector component along the infinite y-axis is non-zero (for oblique incidence in the YZ plane), this needs to be defined also in the environment class (the 2D Green's tensor depends on the corresponding wavevector component[S27]). If the parallel wavevector components of field and environment don't match, an error will be raised.

- `ElectricLineDipole` : An electric line dipole source, infinite along $Y$.

- `MagneticLineDipole` : A magnetic line dipole source, infinite along $Y$.

*3D Illuminations in `tg.env.freespace_3D.inc_field`*

- `PlaneWave` : A plane wave. Supports any kind of polarization and oblique incident angles in the XZ and YZ planes.

- `GaussianParaxial` : A paraxial focused Gaussian beam. Includes E-field divergence correction for tight focus.[S98] Supports any kind of polarization, variable spotsize and focal position, as well as oblique incident angles in the XZ plane.

- `ElectricDipole` : An electric dipole point source.

- `MagneticDipole` : A magnetic dipole point source.

**Classes in `tg.struct2d` and `tg.struct3d`**

The different structure classes available in TorchGDM are depicted in figure 1c-d. 2D structure classes are defined in `tg.struct2d`, where a square lattice surface discretization and several effective models are implemented:

- `StructDiscretized2D` : main surface discretization class

- `StructDiscretizedSquare2D` : surface square lattice discretized structure

- `StructGPM2D` : main 2D GPM class

- `StructTMatrixGPM2D` : GPM from 2D T-Matrix

- `StructMieCylinderGPM2D` : cylindrical GPM from Mie theory

- `StructMieCylinderEffPola2D` : dipole order effective model from Mie theory

3D structures are defined in `tg.struct3d`, containing:

- `StructDiscretized3D` : main volume discretization class

- `StructDiscretizedCubic3D` : cubic lattice volume discretized structure

- `StructDiscretizedHexagonal3D` : hexagonal lattice volume discretized structure

- `StructGPM3D` : main 3D GPM class

- `StructTMatrixGPM3D` : GPM from 3D T-Matrix

- `StructMieSphereGPM3D` : spherical GPM from Mie theory

- `StructMieSphereEffPola3D` : dipole order effective model from Mie theory

All structure classes offer a unified interface for geometry manipulations and plotting. Available methods are:

- `copy()`: Returns a copy of the structure. For creation of multiple identical structures, this accepts as optional argument a list "`positions`" of 3D locations to which the copied structure(s) is (are) moved. Optionally, also "`rotation_angles`" for each of the copies can be specified. If both lists are given, they need to be of equal length.

- `translate(vector)` Returns a copy of the structure, shifted by the input vector.

- `rotate(alpha)` Returns a copy of the structure, rotated clockwise by angle "`alpha`". The rotation axis is given by the optional argument `axis`, which defaults to `"z"`. By default, the location of the rotation axis is the origin, it can be set to another location using the argument `center`.

- `combine(other)`: Combines the structure with an "`other`" structure, that needs to be of same type.

- `plot()`: Plots a 2D projection of the structure

- `plot_contour()`: Plots a 2D projection of the structure's contour

- `plot_3D()`: Plots the structure in 3D

Structure manipulations are also accessible through python addition. For example, combining two structures (of same type!) can be done using the python "+" operator:

```
struct_combined = struct1 + struct2.
```

Translating a structure by a vector $(\Delta x, \Delta y, \Delta z)$ can be done by python adding of a list of 3 elements:

```
struct_shifted = struct + [Δx, Δy, Δz].
```

Finally, structure classes implement a unified interface to access polarizability tensors and self terms. For a complete description of all functions, please see the online documentation. The most relevant methods are:

- `get_polarizability_6x6()`: Returns a set of the full magneto-electric polarizability tensors (6x6) for each meshpoint of the modeled structure. In case of a GPM effective model with $N$ dipole pairs, each structure's non-local global polarizability matrix is of size $(6N \times 6N)$.

- `get_selfterm_6x6()`: Returns a set of the full magneto-electric self-term tensors (6x6) for each meshpoint of the modeled structure.

*Effective polarizability extraction*  Discretized structures support conversion to effective line (2D) or point (3D) polarizability models, using the following methods, which are provided by all discretized structure classes:

- `convert_to_gpm(r_gpm)`,

where `r_gpm` can be either a number (number of dipole pairs automatically distributed inside the structure), or a list of manually defined positions for the effective dipole pairs. Note that if a number is given for `r_gpm`, the extraction is not autodiff compatible because and external clustering algorithm will be used. Conversion to a single effective electric and magnetic dipole pair is possible via:

- `convert_to_effective_polarizability_pair()`.

Both functions require the argument "`wavelengths`", determining all the wavelengths that shall be calculated, as well as the argument "`environment`", to specify the surrounding medium for the effective model.

## 2. Results format

Electromagnetic fields are returned in instances of the `tg.Field` class. Other results are usually returned in dictionaries, for which torchGDM tries to maintain a consistent nomenclature. Observables related to the scattered, total and incident field are returned in dictionary keys carrying the suffixes "sca", "tot" and "inc", respectively. Scattering, extinction and absorption cross widths (2D) and cross sections (3D) are indicated by suffixes "scs", "ecs", "acs".

### `tg.Field` class

The field class contains the complex electric and magnetic fields as well as the positions and optionally surface elements of each location (for definitions on regular grids). It provides various post processing methods, the most relevant are:

- `get_efield()`: Returns the complex electric field at all evaluation positions.

- `get_hfield()`: Returns the complex magnetic field at all evaluation positions.

- `get_efield_intensity()`: Returns the electric field intensity at all evaluation positions.

- `get_hfield_intensity()`: Returns the magnetic field intensity at all evaluation positions.

- `get_poynting()`: Returns the complex Poynting vector at all evaluation positions.

- `get_energy_flux()`: Returns the energy flux by calculating time-averaged Poynting vector at all evaluation positions.

- `get_chirality()`: Returns the near-field chirality at all evaluation positions.

- `get_integrated_efield_intensity()`: Returns the electric field intensity integrated over all evaluation positions. If available this uses the optional surface (or line) elements.

- `get_integrated_hfield_intensity()`: Returns the magnetic field intensity integrated over all evaluation positions. If available this uses the optional surface (or line) elements.

If two `Field` instances are defined on exactly the same positions, the complex fields can be added (or subtracted) using the python "+" (respectively "-") operator:

```
field_superposed = field1 + field2.
```

Furthermore, the `Field` class offers several visualization methods:

- `plot_efield_amplitude()`: 2D plot of the amplitude of (either real or imaginary part) of one Cartesian electric field component at all evaluation positions.

- `plot_efield_vectors()`: 2D quiver plot of the specified projection of the electric vector field real/imaginary part.

- `plot_efield_intensity()`: 2D plot of the electric field intensity at all evaluation positions.

- `plot_efield_vectors3d()`: 3D quiver plot for the electric vector field at all evaluation positions.

- `plot_hfield_amplitude()`: 2D plot of the amplitude of (either real or imaginary part) of one Cartesian magnetic field component at all evaluation positions.

- `plot_hfield_vectors()`: 2D quiver plot of the specified projection of the magnetic vector field real/imaginary part.

- `plot_hfield_intensity()`: 2D plot of the magnetic field intensity at all evaluation positions.

- `plot_hfield_vectors3d()`: 3D quiver plot for the magnetic vector field at all evaluation positions.

- `plot_energy_flux_vectors()`: 2D quiver plot of the time-averaged Poynting vector projection.

- `plot_energy_flux_streamlines()`: Plots the streamlines for the time-averaged Poynting vector.

### 3. Postprocessing

The most relevant postprocessing routines are wrapped into the `Simulation` and `Field` classes (see above). They are also accessible through a functional API in the `tg.postproc` subpackage, which contains the following modules:

- `tg.postproc.crosssect`: Calculation of extinction, absorption and scattering cross sections (in $nm^2$)

- `tg.postproc.fields`: Scattered fields and their gradients in the near-field and far-field zone

- `tg.postproc.multipole`: exact multipole decomposition of the optical response

- `tg.postproc.green`: Green's tensors and LDOS

### 4. Further tools

The subpackage `tg.tools` contains several modules that provide mostly technical tools for tasks such as interpolation, batch processing, geometry transformations or evaluation of pyTorch-compatible special functions. Mie theory helper functions are found in the `tg.tools.mie` module:

- `mie_ab_sphere_3d`: Mie scattering coefficients of a core-shell sphere (3D).

- `mie_ab_cylinder_2d`: Mie scattering coefficients of a core-shell infinite cylinder (2D).

- `mie_crosssections_sphere_3d`: Mie cross sections for a core-shell sphere (3D).

- `mie_crosssections_cylinder_2d`: Mie cross sections for a core-shell infinite cylinder (2D).

Furthermore, tools to work with T-Matrices (via "treams"[S64]) are available in `tg.tools.tmatrix`:

- `cylindrical_wave_source_treams`: Cylindrical wave source as used for conversion of 2D T-Matrices to 2D GPM structures.

- `spherical_wave_source_treams`: Spherical wave source as used for conversion of 3D T-Matrices to GPM structures.

- `convert_tmatrix2D_to_GPM`: convert a set of spectral 2D T-Matrices to a 2D GPM structure.

- `convert_tmatrix3D_to_GPM`: convert a set of spectral 3D T-Matrices to a 3D GPM structure.

As said before, please note that the Mie and T-Matrix modules use "treams",[S64] which does not support automatic differentiation.

Helper functions to create probe positions in 1D (lines) or 2D (planes) are available in the `tg.tools.geometry` module:

- `coordinate_map_1d`: Cartesian equidistant points along a line.

- `coordinate_map_1d_circular`: Cartesian coordinates for points on a circle around the origin in the xz plane with equidistant angular steps.

- `sample_random_circular`: Random points in Cartesian coordinates on an $r = 1$ circle.

- `coordinate_map_2d`: Cartesian equidistant points on rectangular 2d area.

- `coordinate_map_2d_spherical`: Cartesian coordinates of points on a spherical screen around the origin with fixed angular steps.

- `sample_random_spherical`: Random points in Cartesian coordinates on an $r = 1$ sphere.

For a complete description of all tools, modules and functions, please visit the online documentation.

---

* e-mail : pwiecha@laas.fr

[1] Werdehausen, D. *et al.* Modeling Optical Materials at the Single Scatterer Level: The Transition from Homogeneous to Heterogeneous Materials. *Advanced Theory and Simulations* **3**, 2000192 (2020).

[2] Mie, G. Beiträge zur Optik trüber Medien, speziell kolloidaler Metallösungen. *Annalen der Physik* **330**, 377–445 (1908).

[3] Alaee, R., Rockstuhl, C. & Fernandez-Corbaton, I. An elec-tromagnetic multipole expansion beyond the long-wavelength approximation. *Optics Communications* **407**, 17–21 (2018). 1701.00755.

[4] Alaee, R., Rockstuhl, C. & Fernandez-Corbaton, I. Exact Multi-polar Decompositions with Applications in Nanophotonics. *Advanced Optical Materials* **7**, 1800783 (2019).

[5] Majorel, C. *et al.* Generalizing the exact multipole expansion: Density of multipole modes in complex photonic nanostructures.

*Nanophotonics* **11**, 3663–3678 (2022). 2204.13402.

6. Kildishev, A. V., Achouri, K. & Smirnova, D. The Art of Finding the Optimal Scattering Center(s). *Advanced Optical Materials* **13**, 2402787 (2025).

7. Waterman, P. Matrix formulation of electromagnetic scattering. *Proceedings of the IEEE* **53**, 805–812 (1965).

8. Peterson, B. & Ström, S. T Matrix for Electromagnetic Scattering from an Arbitrary Number of Scatterers and Representations of E(3). *Physical Review D* **8**, 3661–3678 (1973).

9. Mishchenko, M. I., Travis, L. D. & Mackowski, D. W. T-matrix method and its applications to electromagnetic scattering by particles: A current perspective. *Journal of Quantitative Spectroscopy and Radiative Transfer* **111**, 1700–1703 (2010).

10. Egel, A. *et al.* SMUTHI: A python package for the simulation of light scattering by multiple particles near or between planar interfaces. *Journal of Quantitative Spectroscopy and Radiative Transfer* **273**, 107846 (2021). 2105.04259.

11. Theobald, D. *et al.* Simulation of light scattering in large, disordered nanostructures using a periodic T-matrix method. *Journal of Quantitative Spectroscopy and Radiative Transfer* **272**, 107802 (2021).

12. Schebarchov, D., Fazel-Najafabadi, A., Le Ru, E. C. & Auguié, B. Multiple scattering of light in nanoparticle assemblies: User guide for the terms program. *Journal of Quantitative Spectroscopy and Radiative Transfer* **284**, 108131 (2022).

13. Mulholland, G. W., Bohren, C. F. & Fuller, K. A. Light Scattering by Agglomerates: Coupled Electric and Magnetic Dipole Method. *Langmuir* **10**, 2533–2546 (1994).

14. Chaumet, P. C. & Rahmani, A. Coupled-dipole method for magnetic and negative-refraction materials. *Journal of Quantitative Spectroscopy and Radiative Transfer* **110**, 22–29 (2009).

15. Patoux, A. *et al.* Polarizabilities of complex individual dielectric or plasmonic nanostructures. *Physical Review B* **101**, 235418 (2020). 1912.04124.

16. Fradkin, I. M., Dyakov, S. A. & Gippius, N. A. Nanoparticle lattices with bases: Fourier modal method and dipole approximation. *Physical Review B* **102**, 045432 (2020).

17. Martin, T. T-matrix method for closely adjacent obstacles. *Journal of Quantitative Spectroscopy and Radiative Transfer* **234**, 40–46 (2019).

18. Capolino, F., Jackson, D. R., Wilton, D. R. & Felsen, L. B. Comparison of Methods for Calculating the Field Excited by a Dipole Near a 2-D Periodic Material. *IEEE Transactions on Antennas and Propagation* **55**, 1644–1655 (2007).

19. Lunnemann, P. & Koenderink, A. F. The local density of optical states of a metasurface. *Scientific Reports* **6**, srep20655 (2016).

20. Kogon, A. J. & Sarris, C. D. FDTD Modeling of Periodic Structures: A Review (2020). 2007.05091.

21. Bertrand, M., Devilez, A., Hugonin, J.-P., Lalanne, P. & Vynck, K. Global polarizability matrix method for efficient modeling of light scattering by dense ensembles of non-spherical particles in stratified media. *JOSA A* **37**, 70–83 (2020). 1907.12823.

22. Lamprianidis, A. G., Rockstuhl, C. & Fernandez-Corbaton, I. Transcending the Rayleigh Hypothesis with multipolar sources distributed across the topological skeleton of a scatterer. *Journal of Quantitative Spectroscopy and Radiative Transfer* **296**, 108455 (2023).

23. Martin, O. J. F., Girard, C. & Dereux, A. Generalized Field Propagator for Electromagnetic Scattering and Light Confinement. *Physical Review Letters* **74**, 526–529 (1995).

24. Girard, C. Near fields in nanostructures. *Reports on Progress in Physics* **68**, 1883–1933 (2005).

25. Mun, J., So, S., Jang, J. & Rho, J. Describing Meta-Atoms Using the Exact Higher-Order Polarizability Tensors. *ACS Photonics* **7**, 1153–1162 (2020).

26. Wiecha, P. R. pyGDM—A python toolkit for full-field electrodynamical simulations and evolutionary optimization of nanostructures. *Computer Physics Communications* **233**, 167–192 (2018).

27. Wiecha, P. R. *et al.* pyGDM – new functionalities and major improvements to the python toolkit for nano-optics full-field simulations. *Computer Physics Communications* **270**, 108142 (2022). 2105.04587.

28. Cazé, A., Pierrat, R. & Carminati, R. Spatial Coherence in Complex Photonic and Plasmonic Systems. *Physical Review Letters* **110**, 063903 (2013).

29. Carminati, R. *et al.* Electromagnetic density of states in complex plasmonic systems. *Surface Science Reports* **70**, 1–41 (2015).

30. Capers, J. R., Boyes, S. J., Hibbins, A. P. & Horsley, S. A. R. Designing the collective non-local responses of metasurfaces. *Communications Physics* **4**, 1–10 (2021).

31. Capers, J. R., Boyes, S. J., Hibbins, A. P. & Horsley, S. A. R. Designing disordered multi-functional metamaterials using the discrete dipole approximation. *New Journal of Physics* **24**, 113035 (2022).

32. Capers, J. R. *et al.* Multiscale design of large and irregular metamaterials. *Physical Review Applied* **21**, 014005 (2024).

33. Colburn, S. & Majumdar, A. Inverse design and flexible parameterization of meta-optics using algorithmic differentiation. *Communications Physics* **4**, 1–11 (2021).

34. Vial, B. & Hao, Y. Open-Source Computational Photonics with Auto Differentiable Topology Optimization. *Mathematics* **10**, 3912 (2022).

35. Wang, C., Chen, N. & Heidrich, W. dO: A Differentiable Engine for Deep Lens Design of Computational Imaging Systems. *IEEE Transactions on Computational Imaging* **8**, 905–916 (2022).

36. Luce, A., Alaee, R., Knorr, F. & Marquardt, F. Merging automatic differentiation and the adjoint method for photonic inverse design. *Machine Learning: Science and Technology* **5**, 025076 (2024).

37. Odom, T. W., You, E.-A. & Sweeney, C. M. Multiscale Plasmonic Nanoparticles and the Inverse Problem. *The Journal of Physical Chemistry Letters* **3**, 2611–2616 (2012).

38. Minkov, M. *et al.* Inverse Design of Photonic Crystals through Automatic Differentiation. *ACS Photonics* **7**, 1729–1741 (2020).

39. Khaireh-Walieh, A. *et al.* A newcomer's guide to deep learning for inverse design in nano-photonics. *Nanophotonics* **12**, 4387–4414 (2023). 2307.08618.

40. Fischbach, J. D. *et al.* A Framework to Compute Resonances Arising from Multiple Scattering. *Advanced Theory and Simulations* **n/a**, 2400989 (2024).

41. Radford, T. W., Wiecha, P. R., Politi, A., Zeimpekis, I. & Muskens, O. L. Inverse Design of Unitary Transmission Matrices in Silicon Photonic Coupled Waveguide Arrays Using a Neural Adjoint Model. *ACS Photonics* **12**, 1480–1493 (2025).

42. Elsawy, M. M. R., Lanteri, S., Duvigneau, R., Fan, J. A. & Genevet, P. Numerical Optimization Methods for Metasurfaces. *Laser & Photonics Reviews* **14**, 1900445 (2020).

43. Lee, D., Chen, W. W., Wang, L., Chan, Y.-C. & Chen, W. Data-Driven Design for Metamaterials and Multiscale Systems: A Review (2023). 2307.05506.

44. So, S., Mun, J., Park, J. & Rho, J. Revisiting the Design Strategies for Metasurfaces: Fundamental Physics, Optimization, and Beyond. *Advanced Materials* 2206399 (2023).

45. Hsu, L., Dupré, M., Ndao, A., Yellowhair, J. & Kanté, B. Local phase method for designing and optimizing metasurface devices. *Optics Express* **25**, 24974–24982 (2017).

46. Majorel, C., Girard, C., Arbouet, A., Muskens, O. L. & Wiecha, P. R. Deep Learning Enabled Strategies for Modeling of Com-

plex Aperiodic Plasmonic Metasurfaces of Arbitrary Size. *ACS Photonics* **9**, 575–585 (2022). 2110.02109.

47 Doicu, A., Wriedt, T. & Eremin, Y. A. *Light Scattering by Systems of Particles.* Springer Series in OPTICAL SCIENCES (Springer, Berlin, Heidelberg, 2006).

48 Hohenester, U. & Trügler, A. MNPBEM – A Matlab toolbox for the simulation of plasmonic nanoparticles. *Computer Physics Communications* **183**, 370–381 (2012).

49 Hohenester, U. Simulating electron energy loss spectroscopy with the MNPBEM toolbox. *Computer Physics Communications* **185**, 1177–1187 (2014).

50 Waxenegger, J., Trügler, A. & Hohenester, U. Plasmonics simulations with the MNPBEM toolbox: Consideration of substrates and layer structures. *Computer Physics Communications* **193**, 138–150 (2015).

51 Hohenester, U., Reichelt, N. & Unger, G. Nanophotonic resonance modes with the nanobem toolbox. *Computer Physics Communications* **276**, 108337 (2022).

52 Schöberl, J. C++ 11 implementation of finite elements in NGSolve. *Institute for analysis and scientific computing, Vienna University of Technology* **30** (2014).

53 Gangl, P., Sturm, K., Neunteufel, M. & Schöberl, J. Fully and semi-automated shape differentiation in NGSolve. *Structural and Multidisciplinary Optimization* **63**, 1579–1607 (2021).

54 Piller, N. & Martin, O. Increasing the performance of the coupled-dipole approximation: A spectral approach. *IEEE Transactions on Antennas and Propagation* **46**, 1126–1137 (1998).

55 Smunev, D. A., Chaumet, P. C. & Yurkin, M. A. Rectangular dipoles in the discrete dipole approximation. *Journal of Quantitative Spectroscopy and Radiative Transfer* **156**, 67–79 (2015).

56 Chaumet, P. C., Sentenac, A. & Rahmani, A. Coupled dipole method for scatterers with large permittivity. *Physical Review E* **70**, 036606 (2004).

57 Chaumet, P. C., Rahmani, A. & Bryant, G. W. Generalization of the coupled dipole method to periodic structures. *Physical Review B* **67**, 165404 (2003).

58 Draine, B. T. The Discrete-Dipole Approximation and its Application to Interstellar Graphite Grains. *Astrophysical Journal* **333**, 848–872 (1988).

59 Draine, B. T. & Flatau, P. J. User Guide for the Discrete Dipole Approximation Code DDSCAT 7.3. *arXiv:1305.6497 [astro-ph, physics:cond-mat, physics:physics]* (2013). 1305.6497.

60 Chaumet, P. C. *et al.* IFDDA, an easy-to-use code for simulating the field scattered by 3D inhomogeneous objects in a stratified medium: Tutorial. *JOSA A* **38**, 1841–1852 (2021).

61 Yurkin, M. A. & Hoekstra, A. G. The discrete-dipole-approximation code ADDA: Capabilities and known limitations. *Journal of Quantitative Spectroscopy and Radiative Transfer* **112**, 2234–2247 (2011).

62 Huntemann, M., Heygster, G. & Hong, G. Discrete dipole approximation simulations on GPUs using OpenCL—Application on cloud ice particles. *Journal of Computational Science* **2**, 262–271 (2011).

63 Muster, A., Abujetas, D. R., Scheffold, F. & Froufe-Pérez, L. S. CoupledElectricMagneticDipoles.jl - Julia modules for coupled electric and magnetic dipoles method for light scattering, and optical forces in three dimensions. *Computer Physics Communications* 109361 (2024).

64 Beutel, D., Fernandez-Corbaton, I. & Rockstuhl, C. *Treams* – a T-matrix-based scattering code for nanophotonics. *Computer Physics Communications* **297**, 109076 (2024).

65 Beutel, D. *A Holistic Framework for Electromagnetic Scattering Simulations Based on the T-matrix Method.* Ph.D. thesis, Karlsruher Institut für Technologie (KIT) (2024).

66 Hugonin, J. P. & Lalanne, P. RETICOLO software for grating analysis (2023). 2101.00901.

67 Kim, Y. *et al.* Meent: Differentiable Electromagnetic Simulator for Machine Learning (2024). 2406.12904.

68 Schubert, M. F. & Hammond, A. M. Fourier modal method for inverse design of metasurface-enhanced micro-LEDs. *Optics Express* **31**, 42945–42960 (2023).

69 Vial, B. Nannos (2022).

70 Oskooi, A. F. *et al.* MEEP: A flexible free-software package for electromagnetic simulations by the FDTD method. *Computer Physics Communications* **181**, 687–702 (2010).

71 Mahlau, Y. *et al.* A flexible framework for large-scale FDTD simulations: Open-source inverse design for 3D nanostructures (2024). 2412.12360.

72 Sersic, I., Tuambilangana, C., Kampfrath, T. & Koenderink, A. F. Magnetoelectric point scattering theory for metamaterial scatterers. *Physical Review B* **83**, 245102 (2011).

73 Girard, C., Weeber, J.-C., Dereux, A., Martin, O. J. F. & Goudonnet, J.-P. Optical magnetic near-field intensities around nanometer-scale surface structures. *Physical Review B* **55**, 16487–16497 (1997).

74 del Hougne, P., Imani, M. F., Diebold, A. V., Horstmeyer, R. & Smith, D. R. Learned Integrated Sensing Pipeline: Reconfigurable Metasurface Transceivers as Trainable Physical Layer in an Artificial Neural Network. *Advanced Science* **7**, 1901913 (2020).

75 Garg, P. *et al.* Inverse-Designed Dispersive Time-Varying Nanostructures. *Advanced Optical Materials* **13**, 2402444 (2025).

76 Giessen, H. & Vogelgesang, R. Glimpsing the Weak Magnetic Field of Light. *Science* **326**, 529–530 (2009).

77 Girard, C., Dujardin, E., Baffou, G. & Quidant, R. Shaping and manipulation of light fields with bottom-up plasmonic structures. *New Journal of Physics* **10**, 105016 (2008).

78 Duan, R. & Rokhlin, V. High-order quadratures for the solution of scattering problems in two dimensions. *Journal of Computational Physics* **228**, 2152–2174 (2009).

79 Loulas, I., Almpanis, E., Tsakmakidis, K. L., Rockstuhl, C. & Zouros, G. P. Electromagnetic Multipole Theory for Two-dimensional Photonics (2024). 2411.05657.

80 Asadova, N. *et al.* T-matrix representation of optical scattering response: Suggestion for a data format. *Journal of Quantitative Spectroscopy and Radiative Transfer* **333**, 109310 (2025).

81 Bohren, C. F. & Huffman, D. R. *Absorption and Scattering of Light by Small Particles* (Wiley, 1998).

82 García-Etxarri, A. *et al.* Strong magnetic response of submicron Silicon particles in the infrared. *Optics Express* **19**, 4815 (2011).

83 Wiecha, P. R. *et al.* Polarization conversion in plasmonic nanoantennas for metasurfaces using structural asymmetry and mode hybridization. *Scientific Reports* **7**, 40906 (2017).

84 Markel, V. A. Extinction, scattering and absorption of electromagnetic waves in the coupled-dipole approximation. *Journal of Quantitative Spectroscopy and Radiative Transfer* **236**, 106611 (2019).

85 Wiecha, P. R., Arbouet, A., Cuche, A., Paillard, V. & Girard, C. Decay rate of magnetic dipoles near nonmagnetic nanostructures. *Physical Review B* **97**, 085411 (2018).

86 Evlyukhin, A. B., Reinhardt, C., Evlyukhin, E. & Chichkov, B. N. Multipole analysis of light scattering by arbitrary-shaped nanoparticles on a plane surface. *Journal of the Optical Society of America B* **30**, 2589 (2013).

87 Evlyukhin, A. B., Fischer, T., Reinhardt, C. & Chichkov, B. N. Optical theorem and multipole scattering of light by arbitrarily shaped nanoparticles. *Physical Review B* **94**, 205434 (2016).

88 Polyanskiy, M. N. Refractiveindex.info database of optical con-

stants. *Scientific Data* **11**, 94 (2024).

89 Pedregosa, F. *et al.* Scikit-learn: Machine Learning in Python. *Journal of Machine Learning Research* **12**, 2825–2830 (2011).

90 Hunter, J. D. Matplotlib: A 2D Graphics Environment. *Computing in Science & Engineering* **9**, 90–95 (2007).

91 Sullivan, C. B. & Kaszynski, A. A. PyVista: 3D plotting and mesh analysis through a streamlined interface for the Visualization Toolkit (VTK). *Journal of Open Source Software* **4**, 1450 (2019).

92 Dong, Z. *et al.* Nanoscale mapping of optically inaccessible bound-states-in-the-continuum. *Light: Science & Applications* **11**, 20 (2022).

93 Abujetas, D. R. *et al.* Brewster quasi bound states in the continuum in all-dielectric metasurfaces from single magnetic-dipole resonance meta-atoms. *Scientific Reports* **9**, 16048 (2019).

94 Gladyshev, S. *et al.* Inverse design of all-dielectric metasurfaces with accidental bound states in the continuum. *Nanophotonics* **12**, 3767–3779 (2023).

95 Gigli, C. *et al.* Fundamental Limitations of Huygens' Metasurfaces for Optical Beam Shaping. *Laser & Photonics Reviews* **15**, 2000448 (2021).

96 Colas des Francs, G. *et al.* Optical Analogy to Electronic Quantum Corrals. *Physical Review Letters* **86**, 4950–4953 (2001).

97 Chicanne, C. *et al.* Imaging the Local Density of States of Optical Corrals. *Physical Review Letters* **88**, 097402 (2002).

98 Novotny, L. & Hecht, B. *Principles of Nano-Optics* (Cambridge University Press, Cambridge ; New York, 2006).