# Peer review of "TorchGDM: A GPU-Accelerated Python Toolkit for Multi-Scale Electromagnetic Scattering with Automatic Differentiation"

_SciPost Physics Codebases, doi:SciPost Phys. Codebases 60 (2025) , SciPost Phys. Codebases 60-r0.56 (2025)_

## Round 1 · Author Response

Dear Editor,

Please find enclosed a revised version of our manuscript "TorchGDM: A GPU-Accelerated Python Toolkit for Multi-Scale Electromagnetic Scattering with Automatic Differentiation" by S. Ponomareva and co-authors. A detailed point-by-point reply to the reviewers' comments, describing also all our changes, can be found below.

We would like to thank the reviewers for their thorough inspection of the manuscript. Their very valuable remarks and constructive suggestions allowed us to improve the manuscript as well as the toolkit itself. We hope that our work is now suitable for publication in SciPost Physics Codebases.

Sincerely, on behalf of all authors, Peter Wiecha

Reviewer 1

We thank the first reviewer for his effort in evaluating our manuscript and for his positive assessment and constructive remarks.

Point (1) - Open source software

Response:
We fully agree with the reviewer on this point. Not only the barrier-free availability is a great plus of open source software. It also allows people to verify and check the code, understand how it works and reuse parts for own software.

Point (2) - Unified solver framework

The torchGDM toolkit is part of this development and combines several approaches that have been found successful in nanophotonics, including T-matrices, effective models (coupled dipoles), and more first-principles based approaches, such as GDM. This is nicely captured in Fig. 3. In my opinion this unifying approach should be stressed more strongly and explicitly, as it seems to suggest that in the future several complementary Maxwell solvers could be combined under a single umbrella. Could torchGDM be used as a generic Maxwell solver framework, which provides interfaces also for other software solutions? A discussion regarding this point would be helpful.

Response:
We thank the reviewer for the suggestion to more explicitly discuss the aspect that torchGDM combines several concepts such as volume discretization, effective dipole models, GPMs and, indirectly, also T-Matrices and Mie theory. We tried to make this clearer in the revised introduction, see also our response to point (4). Concerning the last question whether torchgdm may become a unified solver framework, we fully agree that an umbrella tool, providing a homogeneous interface to various Maxwell solvers, and being capable to interconnect / couple their results, would be an extremely useful software. However, we are not sure if it would make sense to use torchgdm for this aim. Technically, our toolkit is quite specific. It implements frequency domain problems through coupled dipoles and Green's tensors. This naturally allows to mix effective dipoles (and global polarizability matrices) with volume discretizations. Mie models and T-Matrices are only supported indirectly, after conversion to an effective dipoles based description. In conclusion, we don't believe that torchgdm, at least as it is working right now, would be an ideal starting point for such a unified "umbrella framework".

Point (3) - Numerical accuracy / other methods

In this respect, I also suggest expanding the discussion in Sec. 1A regarding other software solutions. I assume that in addition to T-matrices also BEM and DDA could be coupled to torchGDM (as they account for scatterers embedded in free space, contrary to FEM or FDTD). An issue that should be also discussed more rigorously in Sec. 1A is numerical accuracy and stability. For Galerkin based schemes convergence can be demonstrated mathematically (by the way, the authors might also like to refer to the site https://ngsolve.org/), for most other techniques (including GDM) the situation is more complicated and adverse. I suggest expanding the discussion and to be a little more explicit about the strengths and weaknesses of the different approaches.

Response:
Indeed, other software could be easily used to provide field simulations for effective model extractions. We provide an interface to treams for spheres and generic T-matrices, but in principle any Maxwell solver would work. Also this is not limited to frequency domain approaches like DDA or BEM. Also FEM or FDTD could be used for the extraction procedure (field calculations for various illuminations). TorchGDM actually includes a tool that takes illumination fields and scattered fields as input and fits an effective model on these (https://homepages.laas.fr/pwiecha/torchgdm_doc/generated/generated/torchgdm.struct.struct3d.extract_gpm_from_fields.html, admittedly the function has a quite low level interface, we plan to provide in the future an example, how to use external tools like MEEP for the input data to extract a GPM model). Note that there exist various tools to calculate T-matrices with other simulations frameworks. Storing these T-Matrices in the treams dataformat (https://github.com/tfp-photonics/tmatrix_data_format) makes them then directly accessible for torchGDM. We added a paragraph in the section "II. B Extraction of effective models".

We agree that convergence of the GDM / DDA is a critical point. There are various works that propose methods to improve the convergence for different scenarios like high index dielectrics or high aspect ratio structures, but there is no general convergence guarantee. Therefore, for high precision single particle simulations, other software like BEM or FEM may be more adequate. For accurate simulations with many scatterers or for multi-scale simulations, the ideal workflow in torchgdm would be to use another, more stable tool like BEM for the extraction of a very accurate effective GPM model (possibly through a T-Matrix), and subsequently perform an autodiff-capable simulation using such GPMs in torchGDM.

In section 1A, we added a discussion of these limitations including references that discuss possible improvements of the GDM for different scenarios. We also tried to go a bit more into detail concerning the strengths and weaknesses of the other methods. As our original goal was to provide a non-exhaustive, easy-to-digest list of open source software, we tried to keep a balance between the level of detail and the compactness / readability.

Point (4a) - Better introduction of software capabilities

Personally, I would have liked to see before the formalism presented in Sec. 2 some appraisal of what can be done with the present software. What are the typical problems? When is it useful to have automatic differentiation at hand? Where do the authors see room for future developments? I think that such an outlook (and maybe even showing results in the form of some gallery) would help the readers to have a clearer picture of what can be done with the software before entering all the code technicalities.

Response:
We thank the reviewer for these very constructive suggestions. We added at the beginning of the "Examples" section a subsection "Capabilitites" with a new gallery figure (Fig. 5 in the revised text). This figure summarizes the most relevant simulation capabilities (all of them are fully autodiff compatible).
We furthermore added a new figure (Fig. 9 in the revised text) with its describing paragraph to section "III.C Benchmarks", to better highlight the capabilities of the effective models, implemented in torchgdm. This new example demonstrates the fidelity of a global polarizability matrix consisting of 20 dipole pairs, replacing a complicated scatterer that is illuminated by a local source, placed well within the circumscribing circle of the particle. This is not possible with standard T-Matrix method.
In an attempt to make the scope and target problems of the software clearer, we also revised the introduction of torchGDM in section I.A. (just after the overview on open source Maxwell solvers), and we added a short discussion of the current limitations and future developments.

Point (4b) - API descriptions in section 3

Personally I found the code description in Sec. 3 somewhat hard to read, maybe the authors might like to reconsider whether a mere listing of code elements (which can be also found in the documentation) really makes sense.

Response:
After re-reviewing our manuscript ourselves, we fully agree that the technical lists in section 3 are indeed very long and indeed probably not too helpful in the middle of the main text. In the revised manuscript we therefore moved these to a new appendix. In the main text we kept only a very brief presentation of the global package structure, a quick discussion of the main classes and of important technical implementation details. During this restructuring we renamed section II: "Formalism" $\rightarrow$ "Formalism and implementation".

Point (5) - Stratified media

Does torchGDM allow for the consideration of substrates and stratified media (which would be highly beneficial for many applications)?

Response:
Right now, only free space Green's tensors are implemented in torchGDM. However, it is technically easy to add other environments to torchGDM, thanks to its modular software architecture. We demonstrate in an online example how to implement Green's tensors for other environments (https://homepages.laas.fr/pwiecha/torchgdm_doc/auto_gallery/05_extending/ex_04_own_environment.html). This example code implements an environment with a single interface, described by a mirror-dipole approximation. We added a comment in the manuscript's new subsection "III.A Capabilities".

Point (6) - Unit system

Why cgs units?

Response:
We admit that cgs units are somewhat dated and sometimes render unit conversion difficult. We actually use it purely for historical reasons. During the development it allowed us to test against existing in-house implementations of GDM codes. We want to emphasize (as is also stated in the manuscript), that this matters exclusively at the level of the source code. All returned values are either relative values (for instance all field amplitudes are relative to the illumination amplitude), or are converted to SI-compatible units (e.g. scattering cross sections are returned in nm$^2$). In a future update we may convert all the formalism to SI units, but it is not our current priority.

Point (7) - Unclear sentence

The last sentence on page 3 (Note that it has been demonstrated ...) is unclear, the discussion should be expanded.

Response:
We agree that the sentence was unclear. We reformulated it:

Note that automatic differentiation of the LU decomposition algorithm is numerically stable, which has also been demonstrated in the context of the coupled dipoles formalism in recent literature.

Point (8) - Embedding software "Treams"

Why not embedding treams into pyTorch? Could this be done in the future?

Response:
Unfortunately this is not straightforward. Treams is mainly written in cython, all cython code would need to be re-written in pytorch which is significantly more challenging than translating for example numpy to pytorch. Furthermore, the special functions (Bessel, etc.) required for the T-matrix method are not yet available in pytorch (neither in any of the other popular AD frameworks like tensorflow or JAX). These would need to be implemented for automatic differentiation, which is not trivial either. In conclusion, writing an automatic differentiation capable T-Matrix code is a challenging task by itself and beyond our intended scope for torchGDM.

Point (9) - Shape optimization

Can shape optimizations be performed with automatic differentiation? I have the impression that this is somewhat difficult with the GDM method.

Response:
This is indeed difficult, because it is hard to transform a volume discretization on a regular grid into a continuous representation for the particle shape. We are curently testing generative deep learning methods to translate continuous shape parameters into a volume discretization. While this seems to work in principle, we have not managed to achieve really satisfactory results yet. On the other hand, implementing topology optimization using the material at every fixed grid position as an optimization parameter, would be more or less straightforward and would certainly benefit from the automatic differentiation capability (for example to do wavelength dependent optimizations).

Reviewer 2

We thank also the second reviewer for his effort in evaluating our manuscript and for his positive assessment and constructive remarks.

Point (1) - Fully differentiable

The central claim of being "entirely differentiable" is overstated and contradicted by the manuscript's own admission that core functionalities rely on a non-differentiable external library (treams) or use numerical finite differences instead of autograd (for field gradients).
The major current issue of the manuscript lies in contradictory wording about differentiability. The manuscript makes strong claims that the toolkit is "entirely implemented in PyTorch" and thus "any simulation entirely differentiable". These claims are directly contradicted by the authors' own admission that core functionalities for generating effective models from Mie theory or T-Matrices rely on an external, non-differentiable library (“treams”). This dependency breaks the automatic differentiation chain for specific workflows. A similar case is presented for field gradients, which are computed via finite differences instead of AD. I believe that the manuscript could be improved by summarizing these limitations and explaining which potential workflows will be affected. Right now, these limitations are mentioned in fairly remote places of the manuscript instead of in a unified paragraph.
Furthermore, some readers might be interested in a rigorous justification for using numerical finite differences for field gradients, including a quantitative benchmark comparing its performance (speed and memory) and accuracy against a properly formulated AD-based gradient calculation using PyTorch's backward() or forward-mode AD.

Response to "treams" integration:
We thank the reviewer for pointing out that the AD limitations were not clearly enough described. To better discuss these limitations, we added a new subsection II.G "Automatic differentiation limitations", where all non-autodiff compatible parts are listed and their impact on the simulation workflow is briefly discussed.

It is true that the "treams" extensions for Mie solutions and T-matrix conversion are not auto-differentible. This is due to restrictions in the external software, which we cannot easily resolve, as treams is not autodiff capable (see also our response to reviewer 1, point (8)). We did not have the intention to claim that our software is a Mie tool, neither that it is a T-Matrix solver. It seems that we need to emphasize better, that these functions are not "core functionalities" of the toolkit, but rather interfaces to external tools. We therefore also added a clarification in section I.A.7, explaining that the treams parts are not part of torchGDM itself, but rather an interface to the external non-AD tool treams.

Please note, that only the initialization of effective models from these external tools is not AD-capable. From the moment on when the T-Matrix or Mie structure has been converted into an effective model, any operation within torchGDM is fully automatically differentiable.

The conversion of a GDM discretized structure to an effective model is, on the other hand, fully autodifferentiable. Also, the tool to extract an effective model from a list of externally simulated scattered fields is fully differentiable (https://homepages.laas.fr/pwiecha/torchgdm_doc/generated/generated/torchgdm.struct.struct3d.extract_gpm_from_fields.html). The only limitation is, that the locations, where effective dipoles are to be placed inside the original geometry, need to be given specifically for AD support. If the automatic location routine is used, an external clustering algorithm is called, which is not AD-compatible. We added a note to the online documentation as well as to the manuscript. TorchGDM also provides a tool to optimize an existing effective model on a new structure via autodiff, this tool is itself also fully AD-capable (https://homepages.laas.fr/pwiecha/torchgdm_doc/generated/generated/torchgdm.struct.struct3d.optimize_gpm_from_struct.html).

Response to remark on field gradients functionality:
Since field gradients have typically large numbers of both, input parameters and output values (positions, field vector components, different partial derivatives ...), neither forward nor backward AD are ideally suited. In cases with many input locations, PyTorch's "jacobian" function is approximately 200-300 times slower than finite differences evaluation, at a comparable accuracy of the results. Thus, we had initially provided a finite difference based approach, which indeed does not make use of autodiff. However, we want to emphasize, that also the finite differences approach remains fully autodiffable. For instance it is perfectly possible to calculating second order derivatives or wavelength derivatives of finite-differences field gradients.

torch.func "vmap": In principle we agree that the finite differences implementation may be not ideal in some cases. For instance when very strong gradients occur. Also it is indeed unfortunate to not exploit the AD capabilities of torchGDM for this. We therefore invested some major efforts to make the entire field calculation code compatible with torch's new composable function transforms ("torch.func", https://docs.pytorch.org/docs/stable/func.html). We implemented a field-gradient method that makes use of torch.func's vmap auto-vectorization. Even on thousands of evaluation locations, this new implementation is of similar speed compared to finite differences (in our tests we found a factor of around 2 slower runtimes on CPU, and slightly faster timings on GPU). As torch.func is still in beta, we chose to keep also the finite differences method, the used algorithm can now be selected via an optional parameter in the field_gradients function, with autodiff being the default selection. We discuss this in the revised text and compare both methods in the updated online example (https://homepages.laas.fr/pwiecha/torchgdm_doc/auto_gallery/02_usage/ex_05_field_gradients.html).

Note that not all of torchGDM is yet fully compatible with torch.func, but we plan to work on this in the future as vmap would allow for example efficient batch-evaluation of many (small) simulations.

Point (2) - Equation hyperlink error

In equation (10), in the digital PDF version of the manuscript, the "3" in $d^3$ in the denominator is a hyperlink to the literature citations. Is this a LaTeX hyperref package artifact?

Response:
We thank the reviewer for spotting this problem. We don't understand how this has happened, but it seems to have disappeared in the revised manuscript. At least it does not happen anymore with our XeLaTeX compiler.

Point (3) - Periodic simulations: Array scanning method

The manuscript states: "A phased-array scanning method can be used to decompose the dipole field in periodic contributions, but also then, the local light source needs to be sufficiently far from the array". However, the cited Capolino et al. paper (reference 17) actually concludes the opposite. It introduces the "array scanning method" (ASM) and demonstrates that it is more efficient than the direct plane-wave expansion method (PWM) precisely when the dipole source is close to the periodic structure.

Response:
We thank the editor for spotting this incorrect statement. We originally referred at this location only to the work by Lunnemann and Koenderink about the LDOS close to periodic point sources (Scientific Reports 6, 20655, 2016), where a dipolar effective model approximation is used for the periodic structure. Indeed, the method described by Capolino et al. can be applied to arbitrary structures. We corrected the statement and also added two recent references to this topic.

Point (4) - Effective model extraction numerical conditioning:

The description of the GPM inverse problem (Section II.B) could be augmented with a discussion of its numerical conditioning to emphasize the importance of diverse illuminations for particularly ill-posed cases.

Response:
We thank the reviewer for this suggestion. We added an example to the online documentation (https://homepages.laas.fr/pwiecha/torchgdm_doc/auto_gallery/06_benchmarks/ex_10_numerical_conditioning.html), showcasing how the condition number for both, the main simulations as well as for GPM model extraction can be calculated, to assess the numerical stability of the solutions.

However, please note that the conditioning of the GPM inverse problem is not an ambiguous metric to decide whether a useful effective model has been obtained or not. It merely provides an idea about the impact of small illumination changes on the effective model. It does not necessarily provide an insight, whether or not a sufficient number of illuminations has been used. In the online documentation example one can see that while using small numbers of illuminations for GPM extraction is numerically stable (low condition number), the resulting models likely do not reflect all possible illuminations that the structure may receive, and therefore are probably physically poor models. Increasing the number of illuminations first leads to an increase in the condition number, which may be indicating that the inverse problem becomes ill posed. The conditioning improves by further increasing the number of illuminations, which eventually optimizes both, the generality of the physical model and the numerical stability of the (pseudo-) inversion.

We added a sentence to the manuscript, that it is important to verify the fidelity of an effective model before using it, as the numerical stability and physical accuracy can depend on various parameters such as the number of illuminations.

Point (5) - Upper simulation size limit

In section 4.B Benchmarks, the authors mention an upper limit of 10000-15000 dipoles on an RTX 4090. Is there the potential to double this number when halfing the precision at a negligible cost in simulation accuracy?

Response: As of the current version 2.8 (August 2025), PyTorch has only experimental support for ComplexHalf datatype and many important operations such as torch.matmul are not yet natively supported for half precision complex numbers. It would require a significant amount of low-level code development to make half-precision operations work. Anyways, the full inversion scales with $N^2$ in memory requirement, therefore halving the precision would only result in a memory gain of $\sqrt 2$. Also, the accuracy would probably suffer considerably. In conclusion, we do not believe that the effort of implementing half precision support is worth the potential speed gain.

Note that Changing the precision is easily possible by modifying the default values in torchgdm.constants. Switching between single and double precision is readily possible. As soon as half-precision support will be fully available in PyTorch, it will be automatically supported in torchgdm.

Point (6) - Reference 8 problem

The title of reference 8 in the bibliography is not properly rendered (\$T\$).

Response: We thank the reviewer for the thorough reading, good spot. We corrected this issue.

Point (7) - Equation 13 prefactor

Is the first equation in (13) missing a (n_env)$^2$ factor in the denominator?

Response: No, in our unit system the magnetic dipole terms have an additional $1 / n_{\text{env}}^2$ compared to the electric ones (c.f. also equations (14) and (15)).

Other small modifications

Please note that we also implemened a few modifications not directly related to the reviewers' remarks:

  • We added a legend to figure 8b (former figure 7b)
  • We added some labels to figure 15 (former figure 13)
  • We added a reference to a work on cross section calculation (Markel: JQSRT 236, 106611, 2019)
  • We added a reference to "scikit-learn" that is used by the GPM clustering tool.
  • We added a few further references to other relevant open source codes
  • Code: methods of the simulation class that allow to shift or rotate all structures in the simulation
  • Code: several bugfixes

---

## Round 1 · List of Changes

• Extended the "similar software" section, added comments on convergence and typical use cases
  • Detailed summary of "torchGDM" capabilities and limitations
  • Full list of current automatic differentiation limitations
  • Discussion on field gradient calculations
  • Moved detailed API discussion to the Appendix
  • Added a brief discussion of the package structure and main classes
  • Added an overview figure, illustrating the simulation capabilities (new Fig. 5)
  • Added an example on GPM accuracy for large structures (new Fig. 9)

---

## Editorial Decision

published